# 3D-Printed Bioreceptive Tiles of Reaction–Diffusion (Gierer–Meinhardt Model) for Multi-Scale Algal Strains' Passive Immobilization

Yomna K. Abdallah *[ID] and Alberto T. Estévez *[ID]

iBAG-UIC Barcelona, Institute for Biodigital Architecture & Genetics, Universitat Internacional de Catalunya, 08017 Barcelona, Spain
* Correspondence: yomnaabdallah@uic.es (Y.K.A.); estevez@uic.es (A.T.E.)

**Abstract:** The current architecture practice is shifting towards Green Solutions designed, produced, and operated domestically in a self-sufficient decentralized fashion, following the UN sustainability goals. The current study proposes 3D-printed bioreceptive tiles for the passive immobilization of multi-scale-length algal strains from a mixed culture of *Mougeotia* sp., *Oedogonium foveolatum*, *Zygnema* sp., *Microspora* sp., *Spirogyra* sp., and *Pyrocystis fusiformis*. This customized passive immobilization of the chosen algal strains is designed to achieve bioremediation-integrated solutions in architectural applications. The two bioreceptive tiles following the reaction-diffusion, activator-inhibitor Grier–Meinhardt model have different patterns: P1: Polar periodic, and P2: Strip labyrinth, with niche sizes of 3000 μm and 500 μm, respectively. The results revealed that P2 has a higher immobilization capacity for the various strains, particularly *Microspora* sp., achieving a growth rate 1.65% higher than its activated culture density compared to a 1.08% growth rate on P1, followed by *P. fusiformis* with 1.53% on P2 and 1.3% on P1. These results prove the correspondence between the scale and morphology of the strip labyrinth pattern of P2 and the unbranched filamentous and fusiform large unicellular morphology of the immobilized algal strains cells, with an optimum ratio of 0.05% to 0.75% niche to the cell scale. Furthermore, The Mixed Culture method offered an intertwining net that facilitated the entrapment of the various algal strains into the bioreceptive tile.

**Keywords:** passive immobilization; freshwater algae; bioluminescent algae; diatoms; reaction-diffusion; Gierer–Meinhardt model; mixed culture; multi-scale textured surfaces; fractal; 3D-printed tiles; bioremediation; bioreactors; bioactive architecture; sustainability

## 1. Introduction

Bioreceptive surfaces' integration in architectural applications has started to become more active in the past two decades. However, they have always existed in nature. Bioreceptive surfaces offer a host for living organisms to attach to, proliferate on or within, and grow their cultures on. Rocks constitute one example of natural bioreceptive surfaces, which, within their rough texture, provide niches for various micro beings to live and thrive on or inside them. Thus, the current study focuses on passive bioreceptive surfaces. These surfaces exploit their geometrical characteristics to provide a suitable microenvironment for the living bioactive agents that they host without integrating any possible chemical interactions between the hosted cells and the host.

Recently, there are multiple examples of bioreceptive surface applications in the architecture-built environment as architectural facades or elements. Some of them employ only the geometrical characteristics of the surface to provide niches for organisms to inhabit, such as the bioreceptive concrete facades of Marcos Cruz [1] and the bioreceptive tiles created by Mustafa et al., 2021, who experimented with different geometries to test their capacity to host moss [2]. Moreover, Castillo et al., 2021, employed Swarmal fractal patterns in the design of a self-sufficient photobioreactor of *Chlorella* spp. [3]. Another research

practice focuses on adjusting the chemical characteristics of the host, such as in the study conducted by Veeger et al., 2021, who designed the chemical composition of bioreceptive concrete to host moss and different plants [4]. However, this type of bioreceptivity depends on active immobilization, not passive immobilization. Active immobilization depends on the chemical interactions between the host and the immobilized strains to facilitate their immobilization. This chemical interaction can range from adjusting the pH level of the used material composition to catalysis and ion exchange with the hosted strains. Another possibility for active immobilization is to employ algae-based biopolymers such as bioplastics for the immobilization of various algal strains [5], given that microalgae and marine algae are employed to synthesize bioplastic materials [6–8]. Optimally, both the chemical and geometrical compositions of a material should be customized to create an optimized bioreceptive surface. However, the current research focuses mainly on the capacity of the geometrical composition of a bioreceptive surface. The geometric composition of a bioreceptive surface refers to the topological design of its fractal multi-orientation surfaces. These surfaces and topologies form niches and protrusions in various orientations and degrees to increase the bioreceptivity of the surface, offering multiple spaces for the designated immobilized microbial strains. Thus, the current research hypothesis and scope focus on testing the effect of various scale lengths of a bioreceptive pattern texture on its capacity for cell attachment and immobilization of various algal strains. We generated two patterns from one biobehavioral model that can be used solely as bioreceptive surfaces or inside a bioreactor to increase the algal production yield. The main research hypothesis concerns designing a pattern that can be employed successfully as a bioreceptive surface for the passive cell immobilization of various scale strains in a mixed algal culture.

Passive bioreceptive surfaces of passive immobilization systems are employed in diverse industrial and biotechnological applications to exploit the natural tendency of microalgae to attach to surfaces and grow on them [9,10]. There are various types of carriers, which can be active or passive [11]. Passive immobilization is a type of microbial/cellular immobilization method that depends only on the pattern, texture, and topology of the passive bioreceptive surfaces, without including any possible chemical interactions between the bioreceptive surface and the hosted microbial strain. Furthermore, passive immobilization techniques do not include physical effectors (such as the temperature, pH, pressure, or similar) that might trigger irreversible chemical interactions or ion exchange between the bioreceptive surface and the hosted strain [10,12]. This type of passive immobilization is easily reversible; however, it still requires further research and optimization due to many challenges, such as the maintenance and circulation of media and oxygen to keep the cells alive and the management of the effluent in a non-contaminant way. Bioreceptive materials for passive immobilization vary between natural and synthetic. A natural example is loofa sponges, which are non-toxic, non-reactive, cheap, mechanically strong, and stable in long-term cultures. Akhtar et al., 2004 used loofa sponge to immobilize cells of *Chlorella* for the elimination of nickel (II) from aqueous solutions [13]. Some of the literature mentioned the use of synthetic or processed materials such as polyurethane, other plastics, or glass for passive immobilization [14,15] and elsewhere; however, these experiments did not focus enough on the geometrical design effect on algal cells' attachment and the efficiency of the bioreceptive surface for cell immobilization. Thus, the current study focuses on developing the passive immobilization of multi-scale-length algal strains for immobilization on PLA (Polylactic acid), which is a thermoplastic polyester produced from renewable resources and was considered the most popular bioplastic material in 2021 [16]. We aim to 3D-print the bioreceptive tiles while using sophisticated topological bio-geometrical patterns to achieve maximum cell attachment. On the other hand, these patterned tiles could be employed inside bioreactors for maximizing algal culture production. Furthermore, this multi-scale passive-immobilization system of bioreceptive surfaces is designed to be directly applied as architectural facades, walls, or partitions. Which indicates the use of the easiest and most affordable methods for mixed algal culture maintenance, as well as the ease of production of these bioreceptive surfaces as exhibited in the following sections.

These bioreceptive tiles can be used solely, inside bioreactors, or acting as compact bioreactors. Typically, a bioreactor is a closed environment that maintains specific growth conditions to exploit specific biological functions of certain microbial strains [17]. Generally, a bioreactor refers to any manufactured device that supports a biologically active environment [18], or a biochemical process involving organisms or biochemically active substances derived from such organisms, in aerobic or anaerobic conditions [19], and in batch, fed batch or continuous mode. The culturing method varies between suspensions where the microbial cells are submerged in a liquid medium and solid-state culturing where the cells are attached to a surface of a solid medium. Submerged cultures might employ immobilization with a wide variety of methods for cell attachment or entrapment to enhance the attachability of the culture to the bioreactor surfaces, preventing the culture from being flushed out with effluent [20–22]. The current practice of immobilization is limited in scale due to the superficial occupation of the microbes only on the surfaces of the bioreactor vessel. Therefore, this study aims to optimize the bioreceptive tiles for the immobilization of various algal strains on its textured surfaces to maximize the inhabited surface area, and to develop a water retention capacity in these surfaces to simulate the characteristics of a bioreactor.

Recently, bioreactors have been presented in architecture and the built environment at various scales, thanks to the sustainability challenge that has been motivating biodigital architects and designers to propose multi-disciplinarity architectural research and practice. This is to develop architectural systems that perform as bioreactors hosting useful microbial strains for cheap renewable functions, such as bioremediation including $CO_2$ mitigation, and the removal of heavy metals [12,23] and other toxic chemicals, as well as to generate bioelectricity and bioproducts such as oxygen, proteins, enzymes, dyes, etc. A recent study by Satpati and Pal, 2021, used *Cyanobacteria* and *Chlorella ellipsoidea* for biodiesel production, carbon sequestration, and cadmium accumulation [24] as a natural bioremediation method for heavy metal removal. Early attempts to use this integration of bioreactors into architectural systems is found in the Bio lamps Project [25], where a bioluminescent bacterial *Aliivibrio fischeri* strain was cultured inside bioreactors to provide full light for a flat without any electrical fixtures. Another example of algae bioreactor integration into architectural systems can be found in the works of Ecologic Studio [26], where multiple projects of interior design, architectural facades, and pavilions were conducted to integrate bioreactors to grow green algae strains, such as *Chlorella vulgaris,* to consume $CO_2$ and produce oxygen and other by-products, as well as to provoke social environmental responsibility by integrating the users in the process of algae culturing. A more recent application of a bioreactor as a clean renewable electricity generator in an architectural system was proposed in [27], who suggested the employment of the fungal strain *Aspergillus sydowii* NYKA 510 to produce laccase enzyme and generate bioelectricity for domestic use. Another study by Jaafari et al., 2021, proposed bioelectricity production for domestic use by employing the microalgal species *Spirulina Platensis* in a photobioreactor designed following the Diffusion-Limited Aggregation pattern, which is the mathematical logic of growth of this specific algal strain [28]. These studies led to the designing of a methodology and criteria to integrate bioactive systems into the architectural built environment, solving their operational processes, maintenance and exploring their formal design, coherence of their technical aspects, and feasibility of production [29,30]. These bio-integrated architectural practices are establishing an emerging acceptance of bioactive processes as a sustainable, inherent, and intrinsic part of the built environment and normalizing their management and maintenance by average users. The current study is driving forward the promotion of bioreceptive surfaces as compact bioreactors which do not require enclosure, a continuous media supply or effluent management, as well as having easy implementation and management within the built environment.

The integration of bioreactors and/or bioreceptive surfaces in the architectural built environment reflects the complexity in the separate design and operation processes of these biosystems, which has hindered previous attempts in research and practice to combine

these two systems in architectural built environment applications. Hence, the current study offers a bioreceptive surface that can be used solely as a compact non-enclosed bioreactor or inside a bioreactor as a passive immobilization chip as well, depending on the customized topological design of these tiles to avoid the need of enclosure inside a bioreactor. Hence, they can perform sufficiently as customized bioreceptive surfaces with a bioreactor capacity, particularly, in maintaining aqueous culture conditions without continuous suppliers or enclosure.

Thus, two bioreceptive tiles of two different patterns are designed following the same bio-mathematical logic, resembling the growth and proliferation of the tested algal strains. Each pattern is different in terms of its fractal dimension and topological surfaces. These bioreceptive tiles are tested for moisture retention, as well as their passive immobilization capacity of different algal strains with multi-scale lengths and varied morphologies, ranging from unicellular to filamentous. These customized bioreceptive tiles are proposed for application in the architectural built environment as green facades and claddings.

Thus, the main objective of the current study is to prove the relevance and effect of the topological design of a bioreceptive pattern on the passive immobilization capacity and water retention of a bioreceptive surface, through testing of the compatibility between the scale and morphology of the immobilized algal strains and the scale and texture of the bioreceptive pattern. This objective and methodology will enable us to establish a design methodology for customized bioreceptive surfaces, for the customized passive immobilization of any algal strain based on the compatibility of the topological design with the algal strain morphology.

## 2. Results

### 2.1. 3D-Printed Bioreceptive Tiles Following the Reaction–Diffusion Gierer–Meinhardt Model: Pattern 1 and 2

In the proposed model, the simulation field was limited to the proposed size of the bioreceptive tiles of $15 \times 15 \times 0.5$ cm, and to limit the time frame of the simulation process, an initially limited ratio (0.10%) of the initial culture density per strain was used to inform the reaction–diffusion model and the auxiliary CA model. The CA model proposed the starting points of cells' location and distribution in the field which initiated the reaction–diffusion simulation in limited time frames and with a manageable number of agents. Figure 1 exhibits the reached different patterns from the reaction–diffusion simulations of the Gierer–Meinhardt model for pattern 1 and pattern 2, respectively.

Figure 1 exhibits the two generated patterns for the bioreceptive tiles P1 and P2. The first pattern P1 follows a polar, periodic pattern with regular spacing all over the pattern and less sharp maxima. The peak width and the spacing of the peaks are of the same order. This develops wider interstitial spaces (niches or wells that are 3 mm) for capturing the cultivated algal strains. Meanwhile, P2 follows a strip, labyrinth pattern in which more tight interstitial spaces and wells were developed that are 500 μm, thanks to the limited diffusion of the activator that produced the tight stripe formation and produced activated neighbors for the activated cells' occupation/spatial unit ($mm^3$).

These tight niches of the bioreceptive tile P2 required a longer 3D printing time than P1, requiring 36 h in comparison to the 18 h duration of the printing process of P1. Furthermore, the decreased printing speed of 35% was intended to control material deposition for a higher-detail shape fidelity. The PLA filament length used for printing one tile of P2 was 33,283.3 mm of 100.07 g plastic weight, and the material cost was EUR 4.60, while for printing one tile of P1, the consumed filament length was 29,840.4 mm of 89.72 g plastic weight, and the material cost was EUR 4.13. The total energy-consumption cost estimation to produce P2 was EUR 7.056 according to the location of production and the domestic electricity prices for business entities of 0.196 EUR/kWh (GlobalPetrolPrices.com, 26 July 2023), while the total energy consumption cost to produce P1 was EUR 3.528. This reveals that the total cost to produce P2 is 1.5 times higher than the production cost of P1.

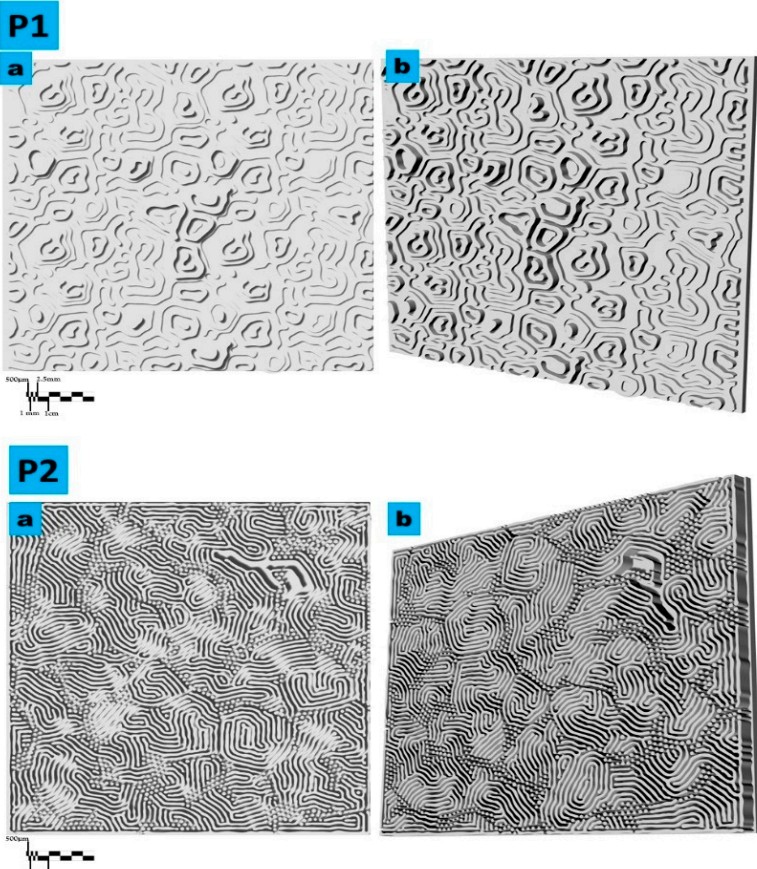

**Figure 1.** The two patterns of the bioreceptive tiles for the passive immobilization of the mixed algae culture: (**P1**) and (**P2**). (**P1**) is a polar periodic pattern employing the following physical parameters for the simulation: the Activator, corresponding to the moisture content, the Inhibitor that corresponds to dry areas/mm$^3$ ranging between 0–1.5%, and the Autocatalytic, which represents a ratio from the initial cell count per each strain. The simulation was run 5 times per strain, after the auxiliary CA model predicted the location of the cells per strain per each simulation iteration, respectively. (**a**) and (**b**) are two different views of the bioreceptive tile (**P1**). (**P2**) is the Strip labyrinth pattern employing the physical parameters of the Activator, corresponding to the moisture content, the Inhibitor, which corresponds to a low moisture content/mm$^3$ ranging between 0.5–5%, and the Autocatalytic, which represents a ratio from the initial cell count per strain. The simulation was run 5 times per strain, after the auxiliary CA model predicted the location of the cells for each simulation iteration, respectively. (**a**), and (**b**) are two different views of the bioreceptive tile (**P2**).

Figure 2 exhibits the printing process of the bioreceptive tile P2, and the two- 3D-printed bioreceptive tiles P1 and P2.

## 2.2. Passive Immobilization of the Multi-Scale Lengths Strains of a Mixed Algal Culture on the 3D-Printed Bioreceptive Tiles: P1 vs. P2

The initial cultivation of each algal strain separately guaranteed their resistance to undergo the test of immobilization on the two different bioreceptive tiles: P1 and P2. Figure 3 exhibits a diagram that represents a comparison between the initial and the activated culture density per algal strain, as well as the immobilized culture density of each of the various algal strains on the bioreceptive tiles P1 and P2, respectively.

As exhibited in Figure 3a, the different algal strains achieved different growth yields. The most potent strain that achieved the highest growth yield compared to its starter culture density was *Pyrocystis fusiformis*, which tripled the density of the culture after 4 weeks of cultivation. This was followed by *Oedogonium foveolatum* and *Mougeotia* sp., achieving

double the density of their starter culture, respectively. The highest culture densities of the immobilized algal strains on P1, as exhibited in Figures 3b and 4, were *Pyrocystis fusiformis* and *Mougeotia* sp., respectively. And on P2, as exhibited in Figures 3c and 5, these were *Microspora* sp., and *Pyrocystis fusiformis*, respectively.

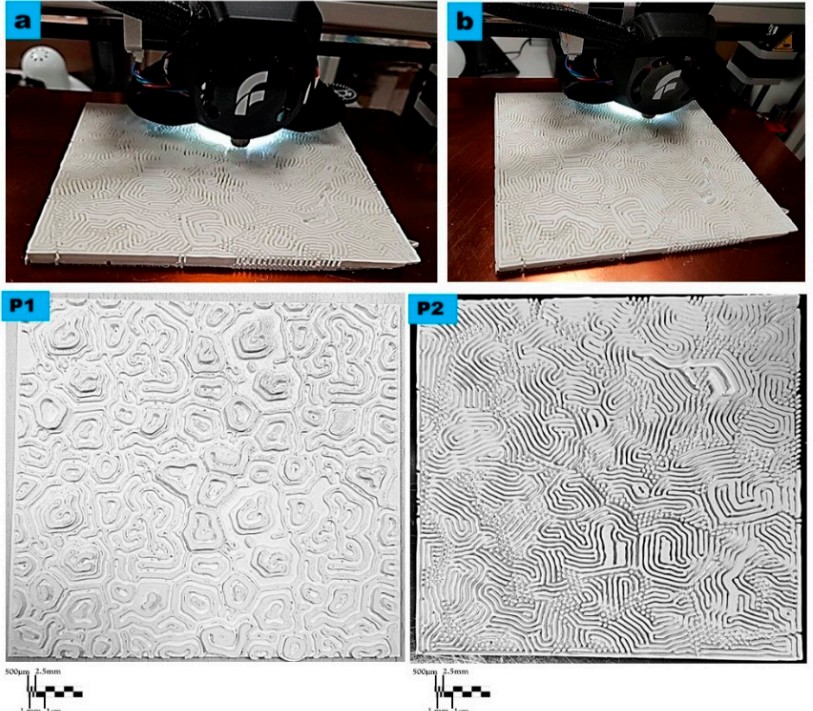

**Figure 2.** The 3D printing process and the 3D-printed tiles (**P1**) and (**P2**): Top: the 3D printing of the bioreceptive tile (**P2**) following the Strip labyrinth, activator-inhibitor, Gierer–Meinhardt model. (**a**) and (**b**) exhibit two views of the printing process that required a slow printing speed of 35% and consumed 36 h of printing to achieve the high shape fidelity of sub-millimeter details. Bottom: the two- 3D-printed bioreceptive tiles (**P1**) and (**P2**) designed for passive immobilization of the mixed algae culture. P1 exhibits the polar periodic pattern with its regular wider niches of 3 mm, while (**P2**) exhibits the strip labyrinth pattern with tight niches of 500 μm, 3D-printed with a high shape fidelity and high-resolution detail.

Figure 4 exhibits the microscopy study results of the bioreceptive tile P1, showing the three levels of examination of the attachability of the algal strains to this polar periodic pattern of the Gierer–Meinhardt activator-inhibitor model.

The results of the microscopy study and culture density estimation from the two samples, P1S1 and P1S2, from the bioreceptive tile P1, exhibited in Figures 3b and 4, revealed that the densest population of the immobilized algal strains on the bioreceptive tile P1 was *Pyrocystis fusiformis,* which increased 1.3 times the inoculated culture density, followed by *Mougeotia*, which achieved a 1.24-fold increase in culture density, and *Microspora*, which increased 1.08 times in comparison to the inoculum culture density.

From this analysis, it can be concluded that the overall affinity of P1 for the immobilization of variant algal strains with variant scale lengths was low to moderate. This is supported by the macro–meso scale microscopy images exhibited in Figure 4, as well as the fact that the most potent immobilized algal strains achieved this anchorage effect due to their morphologies and physical characteristics that facilitated their attachment to the bioreceptive tile P1 despite its relatively wide niches.

However, the bioreceptive tile P2 achieved a higher affinity to attach more algal strains with variant scale lengths, in comparison to P1. Figure 5 exhibits the results of the microscopy study of the algal strains' immobilization on the bioreceptive tile P2 strip

labyrinth pattern of the Gierer–Meinhardt activator-inhibitor model, on three different levels, the macro, the meso, and the micro level.

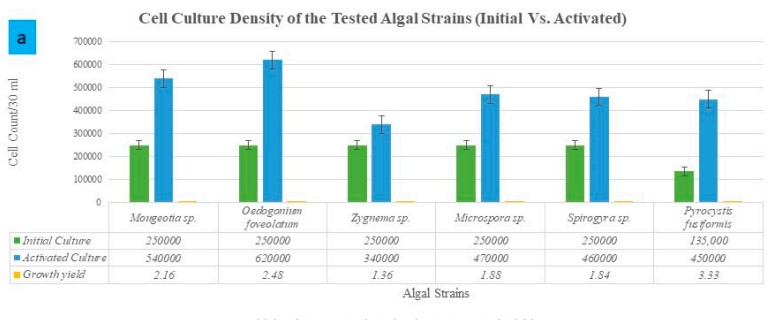

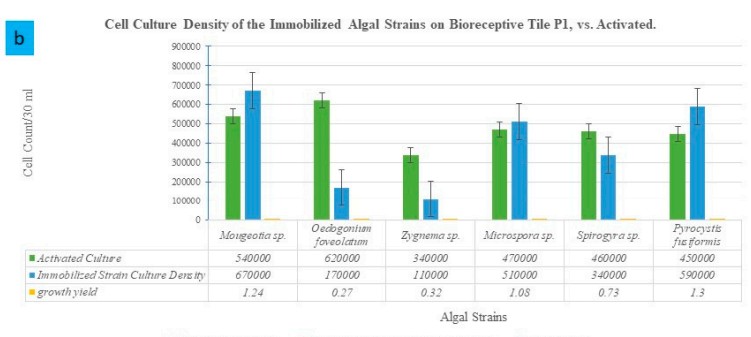

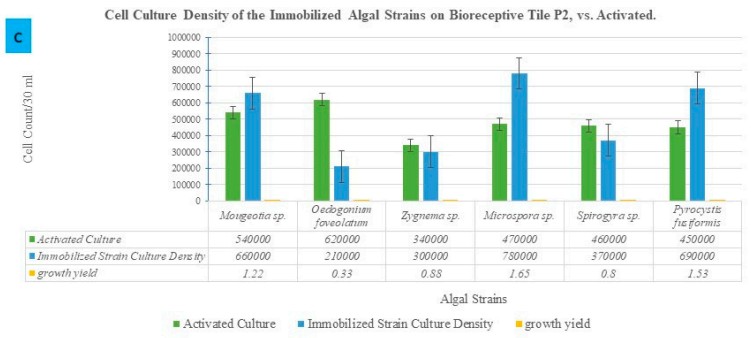

**Figure 3.** Growth rates and cell culture density compared to the initial culture density per strain. (**a**) Cell culture density of the activated cultures of the different algal strains after 4 weeks of cultivating each strain in its optimum growth media and conditions, compared to the starter culture density per strain. (**b**) The estimated cell culture density per each immobilized algal strain on the bioreceptive tile P1, in comparison to the inoculum (activated) culture density per strain. (**c**) The estimated culture density per each immobilized algal strain on the bioreceptive tile P2, in comparison to the inoculum culture density per strain.

The results of the microscopy study and culture density estimation from the two samples, P2S1 and P2S2, from the bioreceptive tile P2, exhibited in Figures 3c and 5, revealed that the densest population of immobilized algal strains on the bioreceptive tile P2 was *Microspora* sp., which had a 1.65-times higher culture density than the inoculum culture density, followed by *Pyrocystis fusiformis,* achieving a 1.53-times larger population. And in third place was *Mougeotia* sp., which achieved a 1.22 higher density compared to the inoculum culture density, while the *Spirogyra* sp. and *Zygnema* sp. immobilized culture densities were slightly decreased from the inoculum culture density, of 80% and 88%, respectively, of the original values. However, they achieved higher densities in comparison to their immobilized cultures on the bioreceptive tile P1. Furthermore, all the tested various algal strains achieved higher densities of their immobilized cultures on the bioreceptive tile P2 in comparison to the bioreceptive tile P1.

This proves the higher capacity of the bioreceptive tile P2, with its tight niches of a strip labyrinth pattern, to capture various scale lengths of the immobilized algal strains ranging from unicellular to filamentous. The bioreceptivity efficiency of P2 in immobilizing multi-scale-length algal strains outweighs the bioreceptivity performance of P1 and compensates for the higher production cost of P2, which was 1.5 times higher than P1. Furthermore, this higher production cost of P2 can be considered as the cost of bioremediation of multiple pollutants and environmental hazards, as well as of an invaluable method of producing byproducts such as enzymes that play a crucial role in industrial applications.

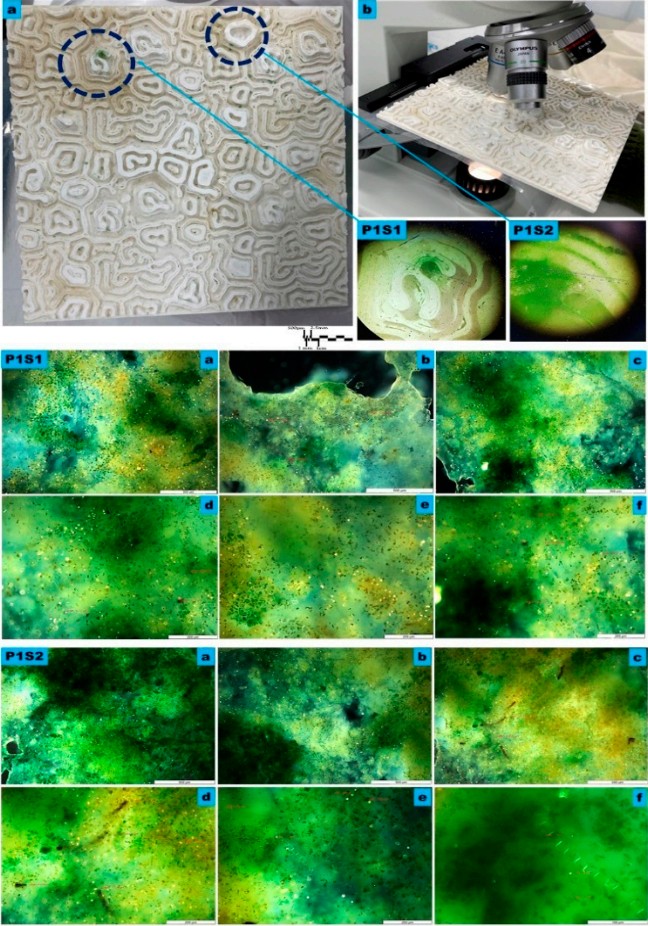

**Figure 4.** The microscopy Study of the macro, meso, and micro levels of the bioreceptive tile **P1**, developed from the Grier–Meinhardt activator-inhibitor model for the polar periodic pattern. (**a**) exhibits the macro level showing the distributed population of the mixed algal culture in a polar pattern starting from different centers, as exhibited within the blue circular dashed lines. (**b**) exhibits the microscopy study under 4X to 8X magnification, used to examine the densest zones of immobilized algal culture, as exhibited in samples **P1S1** and **P1S2** in detail, which were examined with higher magnifications to analyze the various algal strains' attachability to the bioreceptive tile and their different scale-length correspondence to the scale of the developed pattern **P1**. **P1S1** and **P1S2** are microscopic details under 4X, and 8X, respectively. **P1S1** from (**a**) to (**f**) and **P1S2** from (**a**) to (**f**) exhibit the transmission microscopy analysis of samples **P1S1**, and **P1S2**, respectively, at each group; (**a–c**) exhibit the sample under 10X magnification, while (**d–f**) exhibit the sample under 20X. The images exhibit the high density of the immobilized algal strains of *Pyrocystis fusiformis*, *Mougeotia*, and *Microspora*.

### 2.3. Relation between Scale and Morphology of the Bioreceptive Tile Pattern and the Scale and Morphology of the Immobilized Algae Strain

The following, Table 1, exhibits a matrix of all the geometrical parameters affecting the passive immobilization process for each algal strain on each of the bioreceptive tiles.

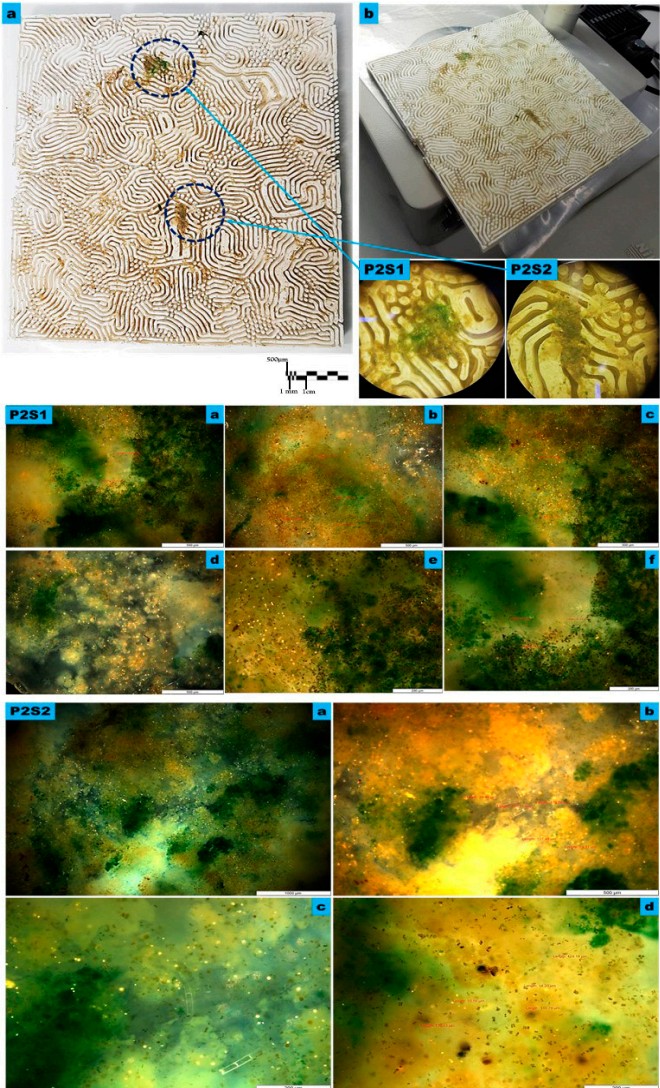

**Figure 5.** The microscopy Study on the macro, meso, and micro levels of the bioreceptive tile **P2**, of the Grier–Meinhardt activator-inhibitor model of the strip labyrinth pattern. (**a**) exhibits the macro level showing the dense immobilized variant algal strains all over the bioreceptive tile **P2**, inhabiting the micro-niches of strip labyrinth pattern, while starting to accumulate in some zones, as highlighted by the blue-dashed-line circles of details (**P2S1**, and **P2S2**). (**b**) exhibits the microscopy study under 4X to 8X magnification of the densest zones of immobilized algal culture, as exhibited in samples P2S1 and P2S2 that were examined with higher magnifications to analyze the various algal strains' attachability to the bioreceptive tile and their different scale-length correspondences to the scale of the developed pattern **P2**. **P2S1** and **P2S2** are microscopic details under 4X magnification. **P2S1** from (**a**) to (**f**) are the transmission microscopy images of sample **P2S1**: (**a**–**d**) exhibit the sample under 10X magnification while (**e**) and (**f**) exhibit the sample under 20X. **P2S2** from (**a**) to (**d**) exhibit the transmission microscopy images of sample **P2S2**: (**a**) exhibits the sample under 5X, and (**b**) exhibits the sample under 10X magnification, while (**c**) and (**d**) exhibit the sample under 20X. The microscopy images of both samples exhibit a high density of the immobilized algal strains *Microspora* sp., followed by *Pyrocystis fusiformis*, *Mougeotia* sp., *Spirogyra* sp., and *Zygnema* sp., respectively.

**Table 1.** Matrix of the ruling parameters of algal-strains immobilization for the bioreceptive tiles, including starter culture density, activated culture density, immobilized culture density on the bioreceptive tile P1, immobilized culture density on the bioreceptive tile P2, growth ratio between starter and activated culture density, growth ratio between immobilized culture density per each tile (P1 and P2), respectively, and starter culture density and activated culture density, respectively (Vs. In = compared to initial or starter culture density; Vs. Act = compared to activated culture density), cell morphology, cell width as indicator of size, relativity of P1 niches to cell width of the strain, P1 hosting capacity, relativity of P2 niches to cell width of strain, and P2 hosting capacity.

| Strain Medium Category | Freshwater Green Filamentous Algae | | | | | | | | Soil-Water Algae | | Marine Water | |
|---|---|---|---|---|---|---|---|---|---|---|---|---|
| Strain Name | *Mougeotia* sp. 2nd in P1 3rd in P2 | | *Oedogonium foveolatum* | | *Zygnema* sp. 5th in P1 5th in P2 | | *Microspora* sp. 3rd in P1 1st in P2 | | *Spirogyra* sp. 4th in P1 4th in P2 | | *Pyrocystis fusiformis* 1st in P1 2nd in P2 | |
| Starter Culture Density/30 mL | 250,000 | Growth Ratio 2.16 | 250,000 | Growth Ratio 2.48 | 250,000 | Growth Ratio 1.36 | 250,000 | Growth Ratio 1.88 | 250,000 | Growth Ratio 1.84 | 135,000 | Growth Ratio 3.33 |
| Activated Culture Density | 540,000 | | 620,000 | | 340,000 | | 470,000 | | 460,000 | | 450,000 | |
| Culture Density on P1 | 670,000 | Vs. In 2.68 / Vs. Act 1.24 | 170,000 | Vs. In 0.68 / Vs. Act 0.27 | 110,000 | Vs. In 0.44 / Vs. Act 0.32 | 510,000 | Vs. In 2.04 / Vs. Act 1.08 | 340,000 | Vs. In 1.36 / Vs. Act 0.73 | 590,000 | Vs. In 4.8 / Vs. Act 1.3 |
| Culture Density on P2 | 660,000 | Vs. In 2.64 / Vs. Act 1.22 | 210,000 | Vs. In 0.84 / Vs. Act 0.33 | 30,0000 | Vs. In 1.2 / Vs. Act 0.88 | 780,000 | Vs. In 3.12 / Vs. Act 1.65 | 370,000 | Vs. In 1.48 / Vs. Act 0.8 | 690,000 | Vs. In 5.1 / Vs. Act 1.53 |
| **Immobilization Ruling parameters** | | | | | | | | | | | | |
| Morphology | Unbranched intertwining filaments | | Unbranched filaments cells wider at one end; occasionally some bulbous cells in between, with rings at the wider end. | | Unbranched short cylindrical cells | | Unbranched filaments with holdfast cells at the end | | Cylindrical cells | | Fusiform shaped, elongated with tapered ends | |
| Cell width | 30 µm | | 45 µm | | 40 µm | | 25 µm | | 90 µm | | 375 µm | |
| Relativity of P1 niches (3000 µm) to cell width of the strain. | 0.01% | | 0.015% | | 0.013% | | 0.008% | | 0.03% | | 0.12% | |
| P1 Hosting Capacity | 100 cells | | 66.6 cells | | 75 cells | | 120 cells | | 33 cells | | 8 cells | |
| Relativity of P2 niches (500 µm) to cell width of strain. | 0.06% | | 0.09% | | 0.08% | | 0.05% | | 0.18% | | 0.75% | |
| P2 Hosting Capacity | 16.6 cells | | 11 cells | | 12.5 cells | | 20 cells | | 5 cells | | 1.3 Cells | |

The immobilization efficiency per each pattern of a bioreceptive tile (P1, and P2) is determined mainly by two major parameters: the cell morphology and size per each algal strain, and the ratio between the niches of the bioreceptive tile and the cell size of each strain separately. The cell size is determined based on the cell width only, excluding the cell length.

The table exhibits the culture density per strain at the four different culturing stages (starter, activated, P1, and P2). The immobilization efficiency per strain was not determined by the culture density number but by the growth or increase ratio compared to the activated culture density per strain.

By analyzing the growth and immobilization behavior of the various algal strains through different stages of the starter culture, activated culture growth, and differential immobilization density per each strain on each of the two bioreceptive tiles, a comparison between the performance of each strain in immobilization efficiency and congruence with each of the bioreceptive tiles' geometry was made, and is presented in Figure 6. This figure exhibits an overall indicator of the geometrical efficiency and adequacy of the multi-scale-length immobilization of each of the bioreceptive tiles, P1 and P2.

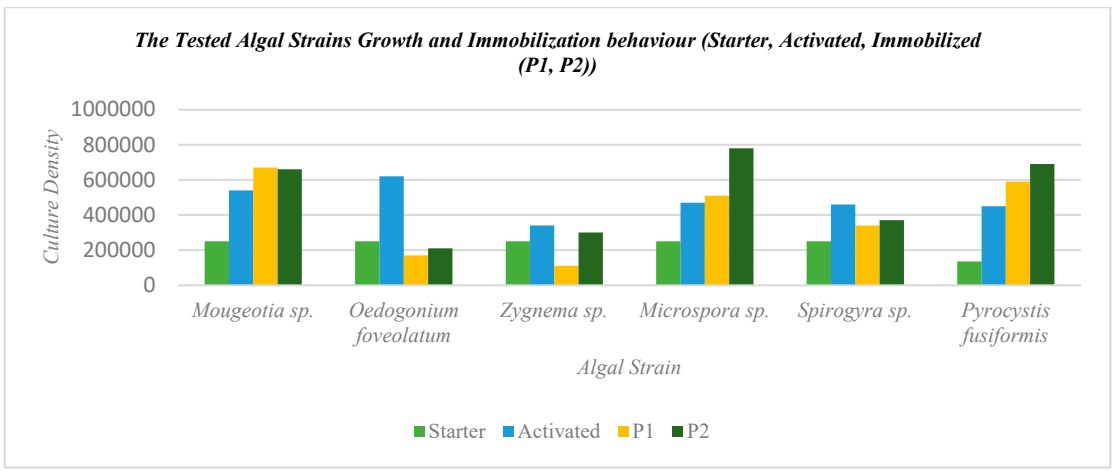

**Figure 6.** Analysis of the tested algal strains' growth and immobilization behavior including the starter or initial culture, the activated culture, and the differential immobilization culture densities on each bioreceptive tile separately, P1 and P2.

From Table 1 and Figure 6, it can be found that the most potent three strains achieving the highest culture density on the bioreceptive tiles, P1 and P2, were *Mougeotia* sp., *Pyrocystis fusiformis, and Microspora* sp., thanks to their morphology that facilitated their attachment to the bioreceptive tiles in both cases with varied ratios. On the other hand, these three strains exhibited regular growth and immobilization behavior on each of the bioreceptive tiles when compared to the growth yield in the activated culture stage. This means that the immobilization process on each of the bioreceptive tiles separately boosted the growth and proliferation of these strains, unlike the other strains of *Spirogyra* sp., *Zygnema* sp., and *Oedogonium faveolatum*, which decreased in culture density in comparison to their activated culture density due to their non-compatible morphology with the two patterns of the bioreceptive tiles. For example, *Oedogonium faveolatum* recorded the least immobilization activity and culture density on each of the bioreceptive tiles P1, and P2, where the immobilized culture density had a drastically decreased ratio compared to the activated culture density, by 0.27% on the bioreceptive tile P1 and 0.33% on P2. This is due to the irregular morphology of the non-even widths of its filaments, with cells that are frequently wider at one end than the other and contain occasionally some bulbous or globular cells in between, with the presence of rings at the wider end [31]. Similarly, the short cylindrical cells of *Zygnema* sp. and *Spirogyra* sp. affected their capacity of anchorage and attachability to each of the bioreceptive tiles. This indicates that the length of the cells

and, consequently, the filaments is contributing to the morphological compatibility of the strain to the bioreceptive tile.

Furthermore, it is proven from Table 1 and Figure 6, that the bioreceptive tile P2, with a strip labyrinth pattern with its tighter niches, achieved an overall higher affinity to immobilize the various tested algal strains, regardless of their category (either freshwater, marine, or soil-water) and morphology, always achieving a higher immobilized culture density than P1 for each strain.

From Table 1, it is exhibited that the best ratio between cell size and bioreceptive niche size was achieved by the bioreceptive tile P2, with a ratio ranging between 0.05% to 0.75%, which indicates the tighter niche size in comparison to P1, in which the ratio between the different strains' cell sizes to their niches' sizes were between 0.008% to 0.12%.

The compatibility of cell morphology with the pattern geometry is as equally important as the niche's size role in cell/filament entrapment and the overall attachability of the bioreceptive tile. Therefore, it can be concluded that the morphology–geometry compatibility contributes equally with the cell size–niche size compatibility to the overall attachability of a bioreceptive immobilization tile.

Finally, it can be concluded from Figure 6 that the mixed culture method contributed to the growth and resistance of the various tested strains, which are from various environments and different growth conditions, such as from freshwater, soil-water, and marine water, which proves the added value of the bioreceptive tiles and the mixed culture growth method in boosting the resistance and consistency of the system in harsh environments with scarce nutrients. Furthermore, the mixed culture that was mainly based on filamentous algae contributed to the immobilization of the various strains due to the lattice mesh that facilitated the mutual support for unicellular and non-filamentous strains to be captured, as well as the stronger anchorage of the mixed culture net to the bioreceptive tile P1, which maintained the moisture entrapment and capture that is mandatory to sustain these aqueous algal strains.

## 3. Discussion

In the current study, the main challenge was to solve the design of the bioreceptive surface without the need for the enclosure inside a controlled environment or for support by a continuous supply of fresh medium and nutrients [32]. Thus, the ruling aspect of this passive immobilization process is the topology and texture of the surface, which allows the attachment and proliferation of the different algal strains' cells [33]. This gives it more application potential in architecture and the built environment, avoiding the complex design requirements of a bioreactor, such as enclosure, physical stability, system circulation, maintenance, and recharging, as has been the case in many recent projects and attempts to integrate algae-cultivation systems in the built environment [34]. This facilitates the freedom of formal design and orientation, as well as the standardization and reproduction of this practice in architecture.

Thus, in the current study, both the passive immobilization and geometry design were proposed to overcome the limitations that a standard bioreactor has and to extend the capacity of a bioreceptive surface as a customized passive immobilization surface that corresponds in scale and morphology to the immobilized strains, with a water retention capacity to maintain the life of these different algal strains. The passive immobilization of the algal strains in the current study was designed for bioremediation purposes to reduce $CO_2$ by consuming it in the active photosynthesis processes of these algal strains, as well as producing oxygen and water vapor [12,35]. This was proposed in [36], which used microtextured chips to immobilize *Chlorella vulgaris* and *Monoraphidium Contortum* for the biodegradation of sulfamethoxazole from wastewater, in order to determine the depth and width of the wells of the textured surface based on the size of the unicellular algal species used and employed 3D printing to translate these fine-scale details in the printed immobilization chip.

*3.1. 3D-Printed Bioreceptive Tiles from The Reaction–Diffusion Gierer–Meinhardt Model: Pattern 1 and 2*

In the current study, the various algal strains' cell sizes and morphologies were considered when designing the pattern's scale of the two bioreceptive tiles for customized passive immobilization. According to the used algal strains, an average size was developed in the offsetting and extrusion of the employed reaction–diffusion activator inhibitor pattern following the morphology and size of the cells of the different algal strains as follows: *Mougeotia strain:* unbranched thalli and intertwining filament morphology with cylindrical cells ranging between 5 and 30 μm in diameter [37]. *Oedogonium:* unbranched filament cells that are frequently wider at one end than the other, with some bulbous or globular cells in-between, and rings at the wider end, with an average cell size of 20–45 μm [37]. *Zygnema cruciatum:* unbranched filaments of short cylindrical cells between 32 and 39 μm in width and 35–50 μm in length, with a diameter of 23.20 μm [38]. *Microspora:* unbranched filaments with cylindrical and holdfast cells at the ends to attach to the substrate, with an average cell size of 25 μm [39]. *Spirogyra:* spiral chloroplast-shaped cells that are usually 118–200 × 240–600 μm, with a width of between 41 and 92 μm, and a length of between 80 and 223 μm [40]. Finally, *Pyrocystis fusiformis:* non-motile, fusiform marine dinoflagellate with cell lengths up to 1 mm [41].

Furthermore, since the five freshwater algal strains are filamentous green algae composed of multiple cells, these filaments were considered when designing the bioreceptive tiles with the two patterns derived from the Gierer–Meinhardt model [42]. Therefore, the XY-offsetting was informed by the average scale lengths of the tested algal strains that were intended to be immobilized on these two bioreceptive tiles P1, and P2 ranging from 500 μm to 1 cm.

The biomathematical model in the current study based on reaction–diffusion [43] offers an abstract logic that describes the gradients of the probability distribution of living algal cells' occupation in a space based on their reaction with the activator and inhibitor concentrations in the medium space [44], which in this case, resembles moisture content that offers an aqueous environment to sustain the various algal strains. This relevance also reflects the cells' occupation and distribution within the topology of a bioreceptive surface over time, informed by the topology simulation for a number of generations of surface occupation generated by an auxiliary cellular automaton simulation [45]. This topology is translated by the reaction–diffusion activator-inhibitor model as niches and protrusions to a textured surface. Thus, the reaction–diffusion model is of deeper relevance to the current case, because the cellular automaton model alone cannot describe the gradients of the algal cells' distribution on a bioreceptive surface for the following reasons: (1) a bioreceptive surface can have various heights and textures more than only two values, that vary in their immobilization capacity; and (2) other parameters regulate the capacity of these protrusions to host living microbial cells, such as their length, width, and material texture (porosity, cohesion, surface finish, particle size, grains, etc.). Thus, all these parameters' gradients are included in the reaction–diffusion model to solve the geometrical composition of the bioreceptive surface [46–51]. Congruently, the application of reaction–diffusion patterns for passive cell immobilization was adopted in a recently published study on the application of a chemical Turing system, engineered to manufacture a porous filter that can be used in water purification [52]. A similar approach was used by Jiang et al., 2005, to immobilize and direct mammalian cell migration through micropatterned surfaces that constrain individual cells to asymmetric geometries; these geometries polarized the morphology of the cells [53].

In the current proposed model, the pattern formation depends on diffusion; the mechanism can only operate if the simulation area is limited. Patterning larger fields indicates a time-consuming lengthy simulation process [44,46–49]. Consequently, in larger fields, the competence to form patterns is lost as the determination of the cells becomes fixed and independent. Thus, in the current study, during the pattern's simulation and development, the limited number of cells were competent only in a certain time window to generate primary organizing regions. Thus, in the current study, the simulation field was

limited to the proposed size of the bioreceptive tiles of $15 \times 15 \times 0.5$ cm, in addition to limiting the time frame of the simulation process by using an initially limited ratio (0.10%) of the initial culture density per strain, while using the same number to inform the auxiliary CA model, which proposed starting points of cells' location and distribution in the field that facilitated the initiation of the reaction–diffusion simulation in limited time frames and with a manageable number of agents.

Thus, in order to generate the two patterns of the bioreceptive tiles, P1 and P2, with their varied topology, P1, which was the polar periodic pattern, employed the moisture content as an activator, with dry areas/mm$^3$ ranging between 0–1.5% as the inhibitor, and the initial cell count per strain as the autocatalytic. Meanwhile, P2, which was the strip labyrinth pattern, employed the moisture content as the activator, the low moisture content/mm$^3$ ranging between 0.5 and 5% as the inhibitor, and the autocatalytic was the ratio of the initial cell count per each strain. The resulting polar, periodic pattern of P1 had regular spacing all over the pattern and less sharp maxima, with a regular order and spacing of the peaks that generated wider niches of 3 mm; meanwhile the strip, labyrinth pattern of P2 had tighter wells that were 500 μm, which required a longer 3D printing time than P1.

*3.2. Multi-Scale-Lengths Algal Strains' Cell Immobilization on the Bioreceptive Tiles P1 and P2*

Freshwater green algae are known for their capacity to perform photosynthesis, consuming carbon dioxide and produce oxygen [54,55], while dinoflagellates are known for their bioluminescence activity [56,57]. Thus, in the current study, five freshwater green algal strains and one bioluminescent dinoflagellate were intended for passive immobilization on bioreceptive tiles for application in architecture, to consume carbon dioxide, and generate oxygen and renewable passive lighting by bioluminescence by designing a bioreceptive surface that could provide niches for algal strains to attach to as well as have the capacity of moist or water retention for the survival of these algal strains. The growth status of the mixed algal culture was guaranteed by the initial cultivation of each strain separately in its optimum growth media and conditions, as recommended by the supplier to boost the resistance and growth of each algal strain culture to undergo the next step of immobilization test on the bioreceptive tiles.

The mixed culture medium was based on water to neutralize the effect of the culture medium on the growth of each strain. The two bioreceptive tiles were suspended in mixed culture to be submerged to facilitate the algal strains' attachment to the tiles while maintaining physical stability and regular exposure to the light–dark cycles.

The sampling method was based on three levels of detecting the algal strains' attachment to each of the bioreceptive tiles. The macro level was focused on detecting the overall attachment pattern of the algae mixed culture to each bioreceptive tile to detect the most populated or dense zones. The meso level was focused on detecting the topology of these dense zones under 4X to 8X magnification to identify the texture and attachment typology and topology of the algal strains on each of the differently patterned bioreceptive tiles, and, to decide on the level and efficiency of attachability per each bioreceptive tile. Finally, the micro level was focused on detecting the ratio of the different algal strains that were immobilized on the bioreceptive tiles, to determine each algal strain culture density and comparing it with the initial and activated culture density per strain, and to measure the effectiveness of the bioreceptive tiles and their biomathematically generated pattern on immobilizing the different algal strains and their viability. Both the meso and micro levels are the ruling criteria to determine the efficiency of the reaction–diffusion different patterns in achieving algal strain immobilization.

The culture-density measurement and counting method employed the Sedgwick Rafter chamber method, because of their compatibility with counting large cells or long chains or colonies where the cell density range is <10,000 cells/mL. Since the current algal culture is a mixed culture with both unicellular and filamentous alga, the Sedgwick Rafter chamber method was useful for the quantification of the different algal strains. Furthermore, the

Sedgwick Rafter chamber was used for microscopy study as well, and specifically using the transmission mode to detect the bioluminescence of *Pyrocystis fusiformis*. This is congruent with the methods conducted in [58,59].

From the microscopy and culture quantification results, the strain that achieved the highest growth yield compared to its starter culture density was *Pyrocystis fusiformis*, with a tripled culture density after 4 weeks of cultivation, followed by *Oedogonium foveolatum* and *Mougeotia* sp., which achieved double the density of their starter culture, respectively. The highest culture densities of immobilized algal strains on P1 were *Pyrocystis fusiformis* and *Mougeotia* sp., respectively. And on P2, they were *Microspora* sp., and *Pyrocystis fusiformis*, respectively.

*Pyrocystis fusiformis* and *Mougeotia* sp. achieved this high attachability to the polar periodic regular pattern of P1 with its relatively wider niches, thanks to their morphological characteristics' compatibility with the pattern of P1, since the interlacing filament morphology of *Mougeotia* facilitated their anchorage in the wells in between the polar pattern peaks as well as around them [37]. Similarly, the non-motile nature of the *Pyrocystis fusiformis* cells and their relatively large size, reaching up to 1 mm, facilitated their anchorage all over the bioreceptive tiles, especially in the wide wells of the polar periodic pattern of the bioreceptive tile P1 [41]. This was similar for the P2 dense immobilized strains. It is obvious that the cell morphology and size of the densest immobilized culture strains are more compatible with the narrower interstitial niches of the topology of P2. The densest immobilized culture was *Microspora* sp., which exploited its unbranched filaments and holdfast cells to anchor to the tighter niches of the strip labyrinth pattern of P2 [39]. This was followed by *Pyrocystis fusiformis*, with its non-motile, fusiform-shaped large-size cells [41,60]. *Mougeotia* sp. came third thanks to its unbranched thalli, intertwining filament morphology, and cylindrical cells [37]. Finally, *Spirogyra* sp. and *Zygnema* sp. came last due to the relatively large size of their cells [38,40].

This proves the hypothesis of the compatibility between the textured bioreceptive surface niches and the designated immobilized strains and its mandatory role in achieving a high cell-immobilization affinity of a bioreceptive surface. This is supported by [61] that proposed cell morphology as a design parameter in the bioengineering of cell–biomaterial surface interactions, suggesting an optimal morphology with a cell aspect ratio (CAR) between 0.2 and 0.4 for both increased cell proliferation and migration.

The immobilization efficiency per each pattern of the bioreceptive tile (P1 and P2) is determined mainly by two major parameters: the cell morphology and size per each algal strain, and the ratio between the niches of the bioreceptive tile topology and the cell size of each strain separately [61–63]. The cell size in this case was determined based on the cell width only, excluding the cell length, since it is a varied parameter per each strain as it should be multiplied by the number of cells in the filament to estimate the total length of the filament for each of the filamentous algal strains tested in this study. The filament length is dependent on the growth phase of each strain culture, since, in a culture, new cells are being generated continuously to generate filaments, which vary in their lengths in each strain culture due to continuous growth. Furthermore, the filament length relativity to the size of the niche at each bioreceptive tile is not congruent in terms of its attachment orientation and morphology, since in this case, the filaments ought to get entangled as fuzzy hairballs, which was the case on both bioreceptive tiles, especially P2 with its tighter wells. Thus, to draw a more measurable insight into the relationship between the cell size and morphology and the niches' scale of the bioreceptive surfaces, only cell width was considered, since all the examined strains were unbranched filamentous algae except for *Pyrocystis fusiformis*, which was unicellular.

The matrix of algal strain immobilization parameters on the bioreceptive tiles exhibits the culture density or cell count per each strain at the four different stages (starter, activated, immobilized on P1, and immobilized on P2). However, since the different strains exhibited varied growth ratios and productivity yields after their activation culturing—which was based on each strain resistance and activity—the immobilization efficiency per each strain

was not determined by the culture density number (cell count), but by the growth or increase ratio compared to the activated culture density per strain. This was conducted to exclude the effect of differential growth capacity and resistance per each strain when cultured in the mixed culture for the immobilization test on the bioreceptive tiles, in the unified culture media, using only tap water without specific nutrients, which would affect the growth capacity per each strain, as well as excluding the influence of the various strains on each other by competence, toxicity, or the parasitic effect. Furthermore, since the two bioreceptive tiles were placed in the same mixed culture pond that already had the culture densities of the different tested strains, the immobilization is evaluated more accurately by the ratio of growth, which indicates the physical attraction force of each biopattern of each bioreceptive tile separately due to the surface tension force that is generated by the proportions of each bioreceptive tile separately, as supported by [64–66].

From the matrix exhibited in Table 1 and Figure 6, the most potent three strains achieving the highest culture density of the bioreceptive tiles, P1 and P2—regardless of their density order—were *Mougeotia* sp., *Pyrocystis fusiformis*, and *Microspora* sp., thanks to their morphology that facilitated their attachment to the bioreceptive tiles in both cases with varied ratios, as explained above. On the other hand, these strains exhibited regular growth and immobilization behavior on each of the bioreceptive tiles when compared to their growth in the activated culture phase, which indicates that the immobilization process on each of the bioreceptive tiles, P1 and P2, separately boosted the growth and proliferation of these strains. This is unlike *Spirogyra* sp., *Zygnema* sp., and *Oedogonium faveolatum*, which decreased in growth as immobilized strains in comparison to their activated culture density, respectively, due to their non-compatible morphology with the two patterns of the bioreceptive tiles. For example, *Oedogonium faveolatum* recorded the least immobilization activity and culture density on each of the bioreceptive tiles where the immobilized culture density had a drastically decreased ratio than the activated culture density by 0.27% on the bioreceptive tile P1 and 0.33% on P2, which was caused by its irregular morphology of the non-even width of its filaments with cells that were frequently wider at one end than the other; occasionally, there were some bulbous or globular cells in between, with the presence of rings at the wider end [32]. This irregular form of its filaments hindered the possibility of anchorage to the wells and niches of each of the bioreceptive tiles, excluding the media stress effect since all the strains were cultivated in the same media based on tap water only. Similarly, the short cylindrical cells of *Zygnema* sp., and *Spirogyra* sp. affected their capacity of anchorage and attachability to each of the bioreceptive tiles. This indicates that the length of the cells and consequently the filaments are still contributing to the morphological compatibility of the strain to the bioreceptive tile, despite not being a fixed parameter that can be compared due to their variance in each strain culture.

Furthermore, it is proven, as exhibited in Table 1 and Figure 6, that the bioreceptive tile P2 of the strip labyrinth pattern with its tighter niches, achieved an overall higher affinity to immobilize the various tested algal strains, from various environments (either freshwater, marine, or soil-water) and morphology, always achieving a higher immobilized culture density than P1 for each strain, which indicates the significant importance of tuning the relativity of the size and scale of the niches of the bioreceptive tile to the size and scale of the algal strain cell/filament that is designated for immobilization. As exhibited in Table 1, the best ratio between cell size and niche size was achieved in the bioreceptive tile P2, with a ratio ranging between 0.05% and 0.75%, resulting from the tighter niche sizes of P2 in comparison to P1 that the ratio between the different strains cell sizes to its niches' sizes were between 0.008% and 0.12%. However, from one perspective, tighter niches or wells indicate a smaller count of possibly hosted cells per niche in the bioreceptive tile as exhibited in Table 1, as the hosting capacity parameter, where P1 should have a higher possibility of hosting more cells per strain. However, the attachability of a bioreceptive tile is ruled more by the entrapment effect that its niches should achieve to capture the algal cells or filaments and immobilize them on the tile, preventing their free motion and consequent loss in and by the aqueous medium. In other words, although larger

niches are supposed to host more cells, due to the big difference between the cell size and the niche size, the entrapment effect is less in wider niches or wells which is the case in P1; this result is congruent with [62] that proved that the algal cell attachment was preferred when the feature size was close to the diameter of the cell attempting to settle. Analyzing the effect of the size ratio between the cell and the niches in P1, this effect was not regular since *Mougeotia* sp., *and Microspora* sp., the second- and the third-most dense immobilized strains on P1, had a smaller ratio between their cells' sizes and the niche sizes of the bioreceptive tile P1, which were 0.01 and 0.008, respectively. However, they achieved better attachability to P1 than *Spirogyra* sp., *Zygnema* sp., and *Oedogonium faveolatum*, which had higher ratios of 0.03, 0.013, and 0.015, while in P2, a more regular relative relation between the ratio of the size of the niches and the size of the cells can be detected, where *Microspora* sp., and *Mougeotia* sp., had approximately the same ratio between their cell sizes and the bioreceptive niches of 0.05%, and 0.75% for *Pyrocystis fusiformis*, being the densest immobilized cultures on P2. However, this was not the case for the least dense cultures such as *Oedogonium faveolatum*, *Zegnema*, and *Spirogyra*, of which their ratios were 0.09%, 0.08%, and 0.18% which were closer to the ratios of *Microspora* and *Mougeotia* on the same bioreceptive tile P2. This is justified by the equal importance of the compatibility of cell morphology with the pattern geometry and the compatibility of niche size with cells/filaments' sizes, which determined the entrapment and overall attachability of the bioreceptive tile. Therefore, it can be concluded that the morphology-geometry compatibility contributes equally to the cell size-niche size compatibility to the overall attachability of a bioreceptive immobilization tile. This justifies the dense immobilization of these three strains on P2, since their unbranched filamentous or fusiform morphologies are compatible with the strip labyrinth pattern of P2 more than the polar periodic pattern of P1.

Finally, it can be concluded that the mixed culture method contributed to the growth and resistance of the various tested strains that are from various environments and with different growth conditions, from freshwater to soil-water and marine water. This proves the added value of the bioreceptive tiles and the mixed culture growth method in boosting the resistance and consistency of the system in harsh environments with scarce nutrients. Furthermore, the mixed culture mainly based on filamentous algae contributed to the immobilization of the various strains due to the lattice mesh effect that facilitated the mutual support for unicellular and non-filamentous strains to be captured as well on the bioreceptive tile P2 with stronger anchorage, as well as maintaining the moisture capture that is mandatory to maintain the life of these aqueous algal strains. This is congruent with [67], which employed the immobilization of mixed algal culture for wastewater treatment upgrading on realistic scales.

## 4. Conclusions

The current work proposes a novel methodology of algal cell immobilization based on the bioreceptive surface geometric design morphology and scale adequacy to the morphology and scale of the designated algal strains' cells. Furthermore, it proposes achieving the water retention capacity of the bioreceptive tiles for the algal cells' immobilization based on the geometrical design of the textured surface as well. This was conducted by proposing the reaction–diffusion activator-inhibitor Gierer–Meinhardt model for the form generation of two bioreceptive tiles, P1 and P2, since the reaction–diffusion mechanism is the most efficient in the formal representation of gradients in any biochemical reaction. These activator-inhibitor Gierer–Meinhardt-based bioreceptive tiles of $15 \times 15 \times 0.5$ cm have two distinct fractal patterns. The two patterns are P1, representing a polar periodic pattern with niches that are 3 mm, and P2, representing a strip labyrinth pattern with niches that are 500 μm. The two bioreceptive tiles were tested for the immobilization of mixed algal culture of multi-scale lengths strains with various morphologies. These were *Mougeotia* sp., *Oedogonium foveolatum*, *Zygnema* sp., *Microspora* sp., *Spirogyra* sp., and *Pyrocystis fusiformis*. The immobilization process utilized neutral media using only tap

water to sustain the mixed culture. Proving the efficiency of the bioreceptive tiles and their customized reaction–diffusion-based patterns in sustaining and immobilizing the various algal strains in stressful growth conditions and media depletion. This was intended since these bioreceptive tiles are proposed for architectural facades or interior wall-cladding applications with minimal maintenance. This was proved by culture density estimation and comparison per each strain along the different stages of the experimental process; from initial culture density, to activated to immobilized culture density on P1 and P2, respectively. The results revealed that the geometrical design of the bioreceptive tile P2 achieved a higher affinity for the various algal strain cells' immobilization. Achieving higher immobilized culture densities of all the tested algal strains in comparison to P1. This was justified by the compatibility between the bioreceptive tile topology and its niches' scale with the immobilized algal strains cells' morphology and scale as well. Furthermore, the strip labyrinth pattern of the bioreceptive tile P2 achieved a higher compatibility with the unbranched filamentous algae morphology, demonstrated by the higher immobilized culture densities achieved by *Microspora* sp., and *Mougeotia* sp., while still being compatible with the large-size fusiform unicellular algae morphology of *Pyrocystis fusiformis.* An optimum ratio between the tested algal strains' cell size and the immobilization niches was identified as 0.05% to 0.75%. This ratio was deducted from analyzing the immobilized algal strains ratio, type, morphology, and performance at each of the bioreceptive tiles. Finally, it was proved that the mixed culture method was a facilitator for immobilizing various algal strains ranging from filamentous to unicellular on the same bioreceptive surface, since the filamentous strains form intertwining meshes and nets to support the anchorage of the various algal strains in the mixed culture to the bioreceptive surface.

## 5. Materials and Methods

This section focuses mainly on the generation of deferential topological surfaces based on translating the biobehavioral logic of reaction–diffusion which is adopted in most biochemical processes in nature [68,69], particularly, in chemotaxis, which is essential for concentration sensing, cell migration, and culture proliferation. Developing the two patterns from the reaction–diffusion model of the Gierer–Meinhardt model was further processed to create variant-scale-length niches within the design of each pattern to enable it to attract various scales of the various algal strains that were intended to be immobilized on these bioreceptive surfaces, as well as to create niches for moist retention to maintain the life of the hosted algal cultures without the need for a liquid media supply. The following section will exhibit the mathematical logic of reaction–diffusion and how it was developed into two different patterns, as well as the 3D printing process of these bioreceptive surfaces, the mixed algal culturing process, and bioreceptive surfaces inoculation. Finally, a microscopy study and analysis were conducted to analyze the affinity of each of the two bioreceptive surfaces to attach variant algal strains tested in this study.

### 5.1. Designing Bioreceptive Surfaces from the Activator-Inhibitor Gierer–Meinhardt Model

Designing a bioreceptive surface requires a differentiable topology to create multiple niches that generate rough-textured surfaces for microalgae to attach to [2]. Thus, designing this textured surface should follow formal and functional specifications that are derived from the morphology of the cultured microbial species, such as the size and morphology of the cells and their aggregation pattern.

In the current study, the capacity of two bio-mathematical patterns that are derived from the same biobehavioral logic were tested in order to identify their varied topologies, and their capacity to host the variant algal species. These biomathematical patterns followed the Reaction–diffusion logic of the Gierer–Meinhardt activator-inhibitor model [47,49], described in Equation (1), to describe the dynamic behavior of proliferating and migrating algal cells influenced by the diffusion of water dots in the simulated medium space of a 15 × 15 × 0.5 cm which is the size of the 3D-printed bioreceptive tile. This converts a stable steady state of a non-spatial system of ordinary differential equations to unstable

due to diffusion leading to diffusion-driven instability. These instabilities cause non-homogenous spatial patterns in two spatial dimensions, giving the patterns of spots, stripes, and labyrinths. In the current study, this variety of patterns was generated from the Gierer–Meinhardt activator-inhibitor model that was parametrized by the physical quantities parameters of the system, which were the moisture content, the algal culture density, and the cellular migration per time unit, that was represented by an auxiliary Conway Game of Life cellular automaton model [70]. Applying the Gierer–Meinhardt model, the identification of the designated morphogens is to create voids or niches to host the various algal strains according to the spatiotemporal predicted concentrations. The used equation (Equation (1)) [47,49] designates *a* as the activator, which stands for moisture content concentration that indicates moist niches attracting the active alive algal cells, while *h* is the inhibitor, which stands for dry surfaces that are not capable of moisture retention, indicating insufficient algal culture attachment.

$$\frac{\partial a}{\partial t} = \rho \frac{a^2}{h} - \mu_a a + D_a \frac{\partial^2 a}{\partial x^2} + \rho_a \tag{1}$$

Equation (1): Reaction–diffusion Gierer–Meinhardt activator-inhibitor model applied in designing the biomathematical patterns of the bioreceptive tiles (General Rule).

$\partial a / \partial t$ describes the change in activator concentration *a* per time unit. This is predicted by an auxiliary Conway Game of Life 3D cellular automaton model that predicts the algal cells' occupation in space, as an indicator of moisture content in each spatial unit (mm$^3$) per time unit (seconds). $\rho$ describes the production rate of algal cells in the total culture density covering the tile space area which depends in a non-linear way on the activator concentration ($a^2$) that is inhibited by the inhibitor (*1/h*). The number of dead cells per time unit is related to the decay rate $\mu_a$ and to the concentration of the activator *a*. The cell migration mode is assumed to occur by diffusion $D_a(\partial^2 a / \partial x^2)$ as described in [49]. These parameters predicted by the 3D cellular automaton model+ the activator-inhibitor model is forming the hypothesis that will be tested by the algal cultures densities and occupation of the bioreceptive tiles after attachment and growth.

These equations of the Gierer–Meinhardt model as well as the auxiliary cellular automaton model were developed by the authors using Rhinoceros 3D+ Grasshopper+ Python. To form the geometrical composition of the two bioreceptive surfaces where this hypothesis of the predicted reaction–diffusion pattern was tested, by the inoculation of the generated patterned bioreceptive tiles and by analyzing their affinity to attach various algal strains with varied lengths scales through the microscopy study.

5.1.1. Two Biopatterns from Gierer–Meinhardt Model: P1: Polar/Periodic and P2: Strip/Labyrinth Patterns

To differentiate the pattern formation process of the two bioreceptive tiles by manipulating the parameters of the developed Gierer–Meinhardt model, two different rules for the reaction–diffusion model are designed. Each of which corresponds to developing a different condition to affect the prediction of algal cell distribution and their related predicted distribution of alive-dead cells in the Conway Game of Life Cellular Automaton model.

The first pattern P1 employs the first rule to generate a polar pattern that is conditioned: if the range of the inhibitor *h* equals the entire field of simulation (which implies the low moist content over all the tile), and if the range of the activator *a* is equal to the total extension, then only a peripheral activation is produced. The interaction between both conditions generates a polar pattern within a field of spatial cells, where the activated regions determine the surrounding region [47,49]. The second rule employed for the P1 design is to generate a periodic pattern. This is conditioned if the range of the inhibitor *h* is lower than the size of the field which is following the *activator-depleted scheme* model. Irregularities of the pattern are caused by the initial fluctuations of the activator-inhibitor size resulting from the predicted distribution of cells from the auxiliary CA model, while maintaining maximum and minimum distance by the size margins of each cell spatial

occupation. Combining the results of these two simulations' diagrams resulted in the first pattern P1 formation.

The second Pattern P2 employed a stripe-like pattern that was conditioned if $\rho a$ the activator-independent production rate saturates at high activator $a$ concentration. Due to this high activator concentration, the peak height cannot increase more. And the spatial extension of a region carrying a high activator concentration increases [71]. Since this is a lateral inhibition mechanism, a stripe-like distribution is formed since in this case each activated cell has an activated neighbor, but also non-activated neighbors are in the vicinity to reduce the inhibitor. This strip-pattern rule is combined with a periodic pattern conditioned by the range of the inhibitor h being smaller than the size of the field, following the activator-depleted scheme model. The resulting pattern was a strip labyrinth pattern.

### 5.1.2. 3D Translation and 3D Printing of the Bioreceptive Tiles: P1, and P2: Z-Offsetting, and Fractal Dimension

To translate the produced 2D reaction–diffusion patterns of the polar periodic bioreceptive tile P1 and the strip labyrinth bioreceptive tile P2, an XY-offsetting and Z- extrusion were applied to each of the two patterns, to provide spatiotemporal reaction–diffusion patterns translated into 3D topologies. The XY-offsetting was informed by the average scale lengths of the tested algal strains that are intended to be immobilized on these two bioreceptive tiles. This was determined by the identification of the cell size and morphology of each of the tested algal strains ranging from 25 μm to 1 cm. The offsets were converted to surfaces that were extruded in the Z direction with an equal numeric value to their width to create proportionate spatial topologies on each bioreceptive tile. Then, the two different 3D patterns P1 and P2 were 3D-printed using Felix Pro Extruder 3D Printer using PLA. The Printing Nozzle size was 0.03 mm, and the printing settings were adjusted to 35% printing speed, 100% flow rate, and 165 °C nozzle temperature. The printing time per tile was between 18 and 36 h, and the printing process was conducted in continuous non-stop mode. After the printing process was complete for each of the tiles, both bioreceptive tiles were sterilized with ethanol 70% and kept in sterile rubber zip bags, to prepare them for the following step of algal inoculation.

### 5.1.3. Multi-Scale Mixed Algal Culture Medium and Inoculation

To prepare the mixed algal culture, $25 \times 10^4$ cells/30 mL of each of five freshwater green algae strains: *Mougeotia* sp., *Oedogonium foveolatum*, *Zygnema* sp., *Microspora* sp., *Spirogyra* sp., and $45 \times 10^4$ cell/100 ml *Pyrocystis fusiformis*., a bioluminescent dinoflagellate, were purchased from (Carolina Biological Supply Company, 2700 York Road, Burlington, NC). *Mougeotia* sp., *Oedogonium foveolatum*, *Zygnema* sp., and *Microspora* sp., were cultivated in customized media (Alga-Gro® Freshwater) provided by the supplier [72], following [73] COMBO medium composition, incubated in 22 °C and 200–400 foot-candles of fluorescent light 18 to 24″ from the culture, for four weeks. While *Spirogyra* sp., was cultivated in Spirogyra Soil-Water medium following the GR+ medium composed of Green House Soil and $CaCO_3$ [74] provided by the supplier in the same temperature, light conditions, and incubation. *Pyrocystis fusiformis* was cultivated in the Bioluminescent Dinoflagellate medium provided by the supplier [75] in 200–400 foot candle of fluorescent light 18 to 24″ from the culture in 12 h light-dark cycles, at 22 °C for four weeks as well.

After the initial cultivation of each algal strain separately, each of them was counted separately using the Sedgwick-Rafter chamber method. Later, all the strains were placed into one sterile pond of $50 \times 30 \times 7$ cm, with a glass cap, cultivated in tape water, in day-light 12 h cycles, and 22 °C. After that, the two PLA 3D-printed and sterile bio receptive tiles P1, and P2 were submerged in the cultivation pond horizontally facing to the top, while being completely submerged within the mixed culture for 60 days with daily mild agitation. The two bioreceptive tiles were picked up from the mixed culture and kept in sterile rubber zip-bags for 24 h while maintaining light and temperature conditions, in preparation for the microscopy study.

### 5.1.4. Culture Density and Microscopy Study

The microscopy study involved two methods of sampling and identification. The first microscopy study was conducted using a Nikon SMZ-2T stereoscope with magnification levels between 4X and 8X for the two- 3D-printed bioreceptive tiles to detect algal strains' attachment all over each of the tiles as well as detect the densest zones of the algal mixed culture on each tile. Consequently, 1mm of each of these dense immobilized cultures was transferred with a sterilized pipet to the Sedgwick Rafter chamber for cell counting [58,76] and further microscopy study. These samples were transferred in separate chambers and labeled as P1S1, and P1S2 from the first bioreceptive tile P1 and P2S1, and P2S2 from the second bioreceptive tile P2. These four samples were studied under the Olympus BX51 transmission microscope with magnification objectives between 5X to 50X for the identification of the immobilized algal strains from the tested algal strains.

The cell-counting method used the Sedgwick Rafter chamber of a volume of 50 mm long, 20 mm wide, and 1 mm deep to calculate the cell density of the dense mixed algal culture. The average was calculated from 10 measurements. The following formula for culture density estimation was used: C = (N × 1000 mm$^3$)/(L × D × W × S), Where: N = number of cells/colonies counted, L = length of transect strip (mm), W = width of transect strip (mm), D = chamber depth (mm), S = number of transects counted [77].

The collected samples (P1S1, P1S2, P2S1, P2S2) were from immobilized cells with no requirement for fixing. Each chamber was filled. The cells were left to settle for 30 min. Once the cells had settled, each chamber was checked at 5 to 50X magnification. The authors intended not to use any dilution methods to avoid any toxicity effect or destruction of cells within the tested mixed culture. For sufficient mixed culture cell counting, the Sedgwick Rafter chamber was divided into a grid of 50 squares long × 20 squares wide. Then, the cells count per strain was made for each square for one or two long transects in the chamber. Counting was performed until the average and standard deviation of counts per square were stable. To perform the culture density estimation for the filamentous algae strains, the area (ABD) for a single cell per each strain was determined, then the filament per each strain was divided by the number of cells in the filament as they appeared in the microscopy images. The following formula was used to calculate the cell density of the filamentous algae sample: ((Mean Area (ABD)) × (Particles per mL)]/(average cell Area (ABD)). The Immobilized culture density per strain on each of the bioreceptive tiles P1 and P2 was compared to the starter culture density, and the activated culture density to identify the effect of immobilization on the culture viability and survival.

**Author Contributions:** Conceptualization, Y.K.A. and A.T.E.; methodology, Y.K.A. and A.T.E.; software, Y.K.A. and A.T.E.; validation, Y.K.A. and A.T.E.; formal analysis, Y.K.A. and A.T.E.; investigation, Y.K.A. and A.T.E.; resources, Y.K.A. and A.T.E.; data curation, Y.K.A. and A.T.E.; writing—original draft preparation, Y.K.A. and A.T.E.; writing—review and editing, Y.K.A. and A.T.E.; visualization, Y.K.A. and A.T.E.; supervision, Y.K.A. and A.T.E.; project administration, Y.K.A. and A.T.E.; funding acquisition, Y.K.A. and A.T.E. All authors have read and agreed to the published version of the manuscript.

**Funding:** This research received no external funding.

**Data Availability Statement:** All the research data is presented in the manuscript; raw data is available upon request to corresponding author.

**Conflicts of Interest:** The authors declare no conflict of interest.

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
