# Peer review of "3D-Printed Bioreceptive Tiles of Reaction–Diffusion (Gierer–Meinhardt Model) for Multi-Scale Algal Strains’ Passive Immobilization"

_buildings, doi:10.3390/buildings13081972_

Round 1
Reviewer 1 Report
The manuscript entitled "3D Printed Bioreceptive Tiles of Reaction-Diffusion (Gierer-Meinhardt model) for Multi-Scale Algal Strains Passive Immobilisation" presents the idea of investigating building materials that would make it easier for objects covered with them to grow algae can be considered to be embedded in the design ideology in architecture called "Designing with Nature." However, is allowing algae to grow better in the walls of buildings the right way to implement "Designing with Nature"? In practice, the opposite is true in buildings inhabited by humans.
Unfortunately, not all algae that could inhabit the surfaces of bioreceptive tile-covered facilities may be friendly to human health; on the contrary, they may become pathogens. The algal environment is also an enabling environment for the growth of pathogenic algae and fungi; for this, it is essential to look at this article from this point of view as well.
The authors have placed the manuscript in a journal dealing with construction techniques, but when analyzing the subject and the content of the manuscript, and the methodology used, I believe that the manuscript is thematically linked to biology and not to construction, which is a technical science. It could be better to publish it in a biological journal. However, since it has been submitted to the Buildings, it should also be assessed from a technical point of view.
The authors wrote that the results of their research would be helpful in the case of cladding buildings with tiles that allow algae to grow, for example, lining bioreactors. The study used PLA material to print the tiles. This is an entirely biodegradable thermoplastic polyester. How do the authors envisage the durability of this material as a building material? How do they envisage this durability when used as a bioreactor lining? Unless they will be disposable bioreactors.
In the developed tile patterns, the authors mainly focused on the type of printed pattern and the width of the cells. From a technical point of view, flat objects, which are containers, and the 3D printed plates are such, with patterns that can be filled with liquid, still have depth and volume and also the ratio of the area of the printed pattern to be filled with liquid, to the total area of the plate. Such data needs to be improved, especially regarding building materials.
The authors wrote that tap water was used to prepare the liquid containing algal cultures. It is well known that the mineral composition of the water determines the growth of different algal species. Why were the minerals and their amounts contained in the water used to prepare the cultures not examined?
The authors compared algae development on two plates with different patterns on their surfaces. In a typical biological study, a reference sample is used as a control from which it is assessed how microorganisms develop in general under the given conditions; for example, in this study, a completely flat plate could have been used, or one with a permanent flat depression on its entire surface.
Furthermore, the manuscript needs to meet the requirements for preparing a text for publication in the scientific journal Buildings MDPI. Please read them at https://www.mdpi.com/journal/buildings/instructions.
I expect the authors to address my allegations in the completed and revised text of the manuscript.
Author Response
Comment.1: " presents the idea of investigating building materials that would make it easier for objects covered with them to grow algae can be considered to be embedded in the design ideology in architecture called "Designing with Nature." However, is allowing algae to grow better in the walls of buildings the right way to implement "Designing with Nature"? In practice, the opposite is true in buildings inhabited by humans.
Answer .1: No, the Manuscript does not present the idea of investigating a building material. it is investigating the effect of geometry and scale of a BIORECEPTIVE SURFACE in the PASSIVE IMMOBILIZATION process of nonpathogenic algal strains. thus, the chemical composition of the material is not a parameter in this process, and there is no chemical interaction between the algal strains and the bioreceptive tiles. the specifically used algal strains are nonpathogenic, on the contrary, they are eco-friendly since they produce oxygen, enzymes, and other useful byproducts, as well as their role in bioremediation which is proved by the literature exhibited throughout the manuscript.
Comment 2: Unfortunately, not all algae that could inhabit the surfaces of bioreceptive tile-covered facilities may be friendly to human health; on the contrary, they may become pathogens. The algal environment is also an enabling environment for the growth of pathogenic algae and fungi; for this, it is essential to look at this article from this point of view as well.
Answer 2: look at the previous comment. furthermore, the study doesn't present a bioreceptive surface for all or any strain of algae, it studies specifically the compatibility between the topological design and scale of the bioreceptive tile and the specific tested five algal strains that are exhibited in the manuscript. again this is passive immobilization not active meaning that the material has no effect and doesn't have any chemical interaction with algae. and again the specific tested algal strains are all nonpathogenic on the contrary they are all proved to have significant roles in the bioremediation and pharmaceuticals industry which is exhibited already in the literature in the manuscript. please avoid generalized judgments we are not speaking about all strains of algae, and the strains that we use are nonpathogenic and we are using PASSIVE IMMOBILIZATION.
Comment 3: The authors have placed the manuscript in a journal dealing with construction techniques, but when analyzing the subject and the content of the manuscript, and the methodology used, I believe that the manuscript is thematically linked to biology and not to construction, which is a technical science. It could be better to publish it in a biological journal. However, since it has been submitted to the Buildings, it should also be assessed from a technical point of view.
Answer 3: should the authors be sorry for submitting to this journal!!!, however, the topic proposes the shift towards green and eco-friendly practices in the built environment. mainly focusing on restoring biodiversity, vegetation and consequently utilizing their photosynthetic capacity to produce oxygen and consume carbon dioxide to reduce their footprint. in addition to the specific bioremediation effects of the utilized algal strains that are already mentioned throughout the manuscript. concerning the technicality of the study, the study focuses specifically on the role of geometry and scale in achieving passi9ve immobilization for bioreceptivity, which can be applied with a wide array of materials and fabrication and construction techniques. Once the bioreceptivity efficiency of the tested topologies and geometries is proved, this methodology and approach can be adopted using any material and any construction technique. thus the protagonist in this study is the geometry.
Comment 4: The authors wrote that the results of their research would be helpful in the case of cladding buildings with tiles that allow algae to grow, for example, lining bioreactors. The study used PLA material to print the tiles. This is an entirely biodegradable thermoplastic polyester. How do the authors envisage the durability of this material as a building material? How do they envisage this durability when used as a bioreactor lining? Unless they will be disposable bioreactors.
Answer 4: look at the previous comment. the protagonist in this study is proving the relationship between geometry and scale of the bioreceptive tiles topology with the hosted algal strains. thus, the used material was only for testing the main hypothesis. however, the idea of disposable bioreceptive tiles is also possible since PLA is biodegradable as the reviewer has mentioned. besides its cost-effectiveness and availability.
Comment 5: In the developed tile patterns, the authors mainly focused on the type of printed pattern and the width of the cells. From a technical point of view, flat objects, which are containers, and the 3D printed plates are such, with patterns that can be filled with liquid, still have depth and volume and also the ratio of the area of the printed pattern to be filled with liquid, to the total area of the plate. Such data needs to be improved, especially regarding building materials.
Answer 5: the authors are aware that there are multiple types and forms of bioreactors and generalized designs of bioreceptive surfaces of moss or other plants. however, the question here is the orientation. is how to design a topology that has water retention capacity through its intricate and reciprocal levels and topologies to maintain water and immobilize algal strains in any orientation. and again the main question is the compatibility between the geometry and scale with the cell morphology and scale. it is a customized design approach and methodology that is the goal of this study. not a generalized geometry application.
Comment 6: The authors wrote that tap water was used to prepare the liquid containing algal cultures. It is well known that the mineral composition of the water determines the growth of different algal species. Why were the minerals and their amounts contained in the water used to prepare the cultures not examined?
Answer 6: no, the cultivation in tap water was a final step after each of the algal strains was cultivated in their optimum media conditions and substrates. tap water generally was used for simplifying the practice for average users that are not specialized microbiologists, so the aim was not to use a synthetic tap water simulated medium, thus the expression of tap water was used.
Comment 7: The authors compared algae development on two plates with different patterns on their surfaces. In a typical biological study, a reference sample is used as a control from which it is assessed how microorganisms develop in general under the given conditions; for example, in this study, a completely flat plate could have been used, or one with a permanent flat depression on its entire surface.
Answer.7: the control in this study is the mixed algal culture pond. a flat plat in this case is the surface of the used pond.
Comment 8: Furthermore, the manuscript needs to meet the requirements for preparing a text for publication in the scientific journal Buildings MDPI. Please read them at https://www.mdpi.com/journal/buildings/instructions.
Answer 8: when the manuscript is accepted, the authors will use the required format.
Comment 9: I expect the authors to address my allegations in the completed and revised text of the manuscript.
Answer 9: the authors have revised the manuscript in light of the reviewer´s comments applying modifications where applicable.
Reviewer 2 Report
This study is very forward-looking, combined with the popular 3D printing technology in recent years, proposed a new algae microbial fixation method, using 3D printing technology to create biosorption tiles with reactive diffusion properties (Gierer-Meinhardt model), and combined with several specific strains to analyze the ratio suitable for each species tile, depending on the algae monarch cell size and their immobilization niches ratio. This not only helps to study the growth distribution of algae under natural conditions, but also helps to find better cultivation methods in experiments, because 3D printing technology can also improve production efficiency and reproducibility, providing the possibility for large-scale applications
Major comment:
Is it a format requirement for this periodical? Why do I think the introduction talks twice about the shortcomings of the previous research and the purpose of this study? The fifth paragraph and beyond seems to be a separate introduction
Minor comment:
1. The 28-line "always" could perhaps be deleted
2. In line 42, "as" might be better replaced by "while"
3. In line 83, "referring to" could perhaps be replaced by "on the other hand"
4. The limitations of the previous study in line 92 should perhaps be placed in the preceding paragraph and the shortcomings of the current study in a paragraph, which focuses on the progressive significance of the study
5. On line 149, delete "per" and the same below
6. The beginning of 3.1 may be brief, reducing the background of the study in the first paragraph to the second paragraph after "thus"
7. At line 435 and beyond, does the formation of the pattern depend solely on diffusion? It is recommended to elaborate.
Author Response
Major comment: Is it a format requirement for this periodical? Why do I think the introduction talks twice about the shortcomings of the previous research and the purpose of this study? The fifth paragraph and beyond seems to be a separate introduction
Answer: when the manuscript is accepted, the required format of this journal will be used. the introduction was revised and modified.
Minor comment:
- The 28-line "always" could perhaps be deleted
- In line 42, "as" might be better replaced by "while"
- In line 83, "referring to" could perhaps be replaced by "on the other hand"
- The limitations of the previous study in line 92 should perhaps be placed in the preceding paragraph and the shortcomings of the current study in a paragraph, which focuses on the progressive significance of the study
- On line 149, delete "per" and the same below
- The beginning of 3.1 may be brief, reducing the background of the study in the first paragraph to the second paragraph after "thus¨.
Answer 1,2,3,4,5,6: the manuscript was revised extensively as well as revised for style and language, and the modifications were applied where applicable without hindering the required meaning.
7. At line 435 and beyond, does the formation of the pattern depend solely on diffusion? It is recommended to elaborate.
Answer 7: pattern formation was further elaborated in the discussion section.
Reviewer 3 Report
Dear Author,
You have prepared this article in good manner. Thanks for choosing this Journal.
Author Response
Comment: You have prepared this article in good manner. Thanks for choosing this Journal.
answer: thank you
Round 2
Reviewer 1 Report
Comment.1: " presents the idea of investigating building materials that would make it easier for objects covered with them to grow algae can be considered to be embedded in the design ideology in architecture called "Designing with Nature." However, is allowing algae to grow better in the walls of buildings the right way to implement "Designing with Nature"? In practice, the opposite is true in buildings inhabited by humans.
Answer .1: No, the Manuscript does not present the idea of investigating a building material. it is investigating the effect of geometry and scale of a BIORECEPTIVE SURFACE in the PASSIVE IMMOBILIZATION process of nonpathogenic algal strains. thus, the chemical composition of the material is not a parameter in this process, and there is no chemical interaction between the algal strains and the bioreceptive tiles. the specifically used algal strains are nonpathogenic, on the contrary, they are eco-friendly since they produce oxygen, enzymes, and other useful byproducts, as well as their role in bioremediation which is proved by the literature exhibited throughout the manuscript.
Comment 2: Unfortunately, not all algae that could inhabit the surfaces of bioreceptive tile-covered facilities may be friendly to human health; on the contrary, they may become pathogens. The algal environment is also an enabling environment for the growth of pathogenic algae and fungi; for this, it is essential to look at this article from this point of view as well.
Answer 2: look at the previous comment. furthermore, the study doesn't present a bioreceptive surface for all or any strain of algae, it studies specifically the compatibility between the topological design and scale of the bioreceptive tile and the specific tested five algal strains that are exhibited in the manuscript. again this is passive immobilization not active meaning that the material has no effect and doesn't have any chemical interaction with algae. and again the specific tested algal strains are all nonpathogenic on the contrary they are all proved to have significant roles in the bioremediation and pharmaceuticals industry which is exhibited already in the literature in the manuscript. please avoid generalized judgments we are not speaking about all strains of algae, and the strains that we use are nonpathogenic and we are using PASSIVE IMMOBILIZATION.
Reviewer's comment
My two comments (1 and 2) on the manuscript were guided by the approximation of its content to the theme of the journal Buildings and to construction as an art, science, and technology. Builders are not interested in whether the immobilisation of algae on the surface of the material they may be using is passive or active and whether the research involved beneficial algae for human health when they have concerns that also the non-beneficial and toxic ones may also develop, so hence were my comments. I found some attempts to clarify this in the revised manuscript, albeit in a non-literal and satisfactory sense.
Comment 3: The authors have placed the manuscript in a journal dealing with construction techniques, but when analyzing the subject and the content of the manuscript, and the methodology used, I believe that the manuscript is thematically linked to biology and not to construction, which is a technical science. It could be better to publish it in a biological journal. However, since it has been submitted to the Buildings, it should also be assessed from a technical point of view.
Answer 3: should the authors be sorry for submitting to this journal!!!, however, the topic proposes the shift towards green and eco-friendly practices in the built environment. mainly focusing on restoring biodiversity, vegetation and consequently utilizing their photosynthetic capacity to produce oxygen and consume carbon dioxide to reduce their footprint. in addition to the specific bioremediation effects of the utilized algal strains that are already mentioned throughout the manuscript. concerning the technicality of the study, the study focuses specifically on the role of geometry and scale in achieving passi9ve immobilization for bioreceptivity, which can be applied with a wide array of materials and fabrication and construction techniques. Once the bioreceptivity efficiency of the tested topologies and geometries is proved, this methodology and approach can be adopted using any material and any construction technique. thus the protagonist in this study is the geometry.
Reviewer's comment
The authors do not have to apologise to anyone for submitting a manuscript to the journal Buildings. It is up to the reviewer and the Journal Editor to judge whether or not the manuscript, thematically in its present form, falls within the scope of this journal.
Comment 4: The authors wrote that the results of their research would be helpful in the case of cladding buildings with tiles that allow algae to grow, for example, lining bioreactors. The study used PLA material to print the tiles. This is an entirely biodegradable thermoplastic polyester. How do the authors envisage the durability of this material as a building material? How do they envisage this durability when used as a bioreactor lining? Unless they will be disposable bioreactors.
Answer 4: look at the previous comment. the protagonist in this study is proving the relationship between geometry and scale of the bioreceptive tiles topology with the hosted algal strains. thus, the used material was only for testing the main hypothesis. however, the idea of disposable bioreceptive tiles is also possible since PLA is biodegradable as the reviewer has mentioned. besides its cost-effectiveness and availability.
Reviewer's comment
This answer satisfied the reviewer.
Comment 5: In the developed tile patterns, the authors mainly focused on the type of printed pattern and the width of the cells. From a technical point of view, flat objects, which are containers, and the 3D printed plates are such, with patterns that can be filled with liquid, still have depth and volume and also the ratio of the area of the printed pattern to be filled with liquid, to the total area of the plate. Such data needs to be improved, especially regarding building materials.
Answer 5: the authors are aware that there are multiple types and forms of bioreactors and generalized designs of bioreceptive surfaces of moss or other plants. however, the question here is the orientation. is how to design a topology that has water retention capacity through its intricate and reciprocal levels and topologies to maintain water and immobilize algal strains in any orientation. and again the main question is the compatibility between the geometry and scale with the cell morphology and scale. it is a customized design approach and methodology that is the goal of this study. not a generalized geometry application.
Reviewer's comment
The authors use the terms 'geometry' and "topology." However, these terms refer to patterns or spatial elements - 3D, and since the pattern on the plate was printed and therefore also had a third dimension - depth, which I asked about and about which I found no mention in the text of the manuscript and the authors' response to my comments. After all, the depth of the printed pattern also affects its surface area from a geometrical and topographical point of view and the water capacity of the printed pattern. Because it is a spatial pattern, the dept can also affect the amount of water held and the immobilisation of algal strains in any orientation. In addition, there is no information as to the ratios of the area of the two patterns to the total area of the tiles, although the dimensions: of length, width, and thickness of the tiles are given.
Comment 6: The authors wrote that tap water was used to prepare the liquid containing algal cultures. It is well known that the mineral composition of the water determines the growth of different algal species. Why were the minerals and their amounts contained in the water used to prepare the cultures not examined?
Answer 6: no, the cultivation in tap water was a final step after each of the algal strains was cultivated in their optimum media conditions and substrates. tap water generally was used for simplifying the practice for average users that are not specialized microbiologists, so the aim was not to use a synthetic tap water simulated medium, thus the expression of tap water was used.
Reviewer's comment
Then please write in the text of the manuscript that the study used "expressed tap water", but even this "expression tap water" had some mineral composition?
Comment 7: The authors compared algae development on two plates with different patterns on their surfaces. In a typical biological study, a reference sample is used as a control from which it is assessed how microorganisms develop in general under the given conditions; for example, in this study, a completely flat plate could have been used, or one with a permanent flat depression on its entire surface.
Answer.7: the control in this study is the mixed algal culture pond. a flat plat in this case is the surface of the used pond.
Reviewer's comment
Interesting way to control the research, and where to find the results for comparison?
Comment 8: Furthermore, the manuscript needs to meet the requirements for preparing a text for publication in the scientific journal Buildings MDPI. Please read them at https://www.mdpi.com/journal/buildings/instructions.
Answer 8: when the manuscript is accepted, the authors will use the required format.
Reviewer's comment
The required format applies to manuscripts at the level of submission and not acceptance. The reviewer assesses the content and form of the manuscript that will be published, not just the content, and the form will be reworked as the content is accepted. The fact that a journal editor decided to submit a manuscript for review in a form incompatible with the publisher's form can only be due to the editor's courtesy, but this does not mean that the reviewer's comments can be ignored.
Comment 9: I expect the authors to address my allegations in the completed and revised text of the manuscript.
Answer 9: the authors have revised the manuscript in light of the reviewer´s comments applying modifications where applicable.
Reviewer's comment
According to the rules for responding to reviewer comments, authors should indicate the line numbers in the revised manuscript where they have made changes under the reviewer's comments.
Additional comments
In addition, a misspelling of platelet sizes in the revised manuscript was noted (lines: 158 and 159)
should be either: 15.0 cm x 15.0 cm x 0.5 cm or as the capacity resulting from the multiplication of the given values, i.e., 112.5 cm3. The notation "cm^3" is used as a command for calculations in an Excel spreadsheet and has nothing to do with the volumes and values presented in the manuscript's text.
Final comment
The reviewer has commented on the current content and form of the manuscript, and the authors are left with the final decision on what to change or add to it, bearing in mind the reviewer's comments.
Author Response
Reviewer's comment 1-Round 2: My two comments (1 and 2) on the manuscript were guided by the approximation of its content to the theme of the journal Buildings and to construction as an art, science, and technology. Builders are not interested in whether the immobilization of algae on the surface of the material they may be using is passive or active and whether the research involved beneficial algae for human health when they have concerns that also the non-beneficial and toxic ones may also develop, so hence were my comments. I found some attempts to clarify this in the revised manuscript, albeit in a non-literal and satisfactory sense.
Answer 1: the authors thank the reviewer for their further explanation. however, the authors insist on their approach of focusing on geometry and scale as the main protagonist of the study and their role in passive immobilization of algal strains that are now mandatory in the green and eco-friendly shift of the construction industry to host and boost biodiversity and embedded bioactive systems for the mandatory bioremediation of carbon dioxide harsh effects on climate change. Indeed, the topic doesn't address the conventional classic understanding of the construction industry, that have stunted growth since the sixties reproducing concrete boxes and depleting resources while leaving no place for green life and biodiversity to grow and thrive which our planet currently is paying the price for this strict boxed choice. thus, our study presents what is needed the most in the construction industry and technology; to care about biodiversity and useful algal strains and what they can serve in producing oxygen and useful byproducts while performing a significant function of bioremediation. Trying to mitigate the effects of strict conventional construction methods that have led our planet to the current critical situation. Thus, based on this approach and understanding the authors have explained this approach throughout the manuscript while maintaining the focus on the specific scope of this experimental study ¨the geometry and scale compatibility for passive immobilization of useful algal strains¨ for developing bioreceptive tiles that is an embedded bioactive system in the built environment for bioremediation purposes.
Reviewer comment 2- Round 2: The authors do not have to apologize to anyone for submitting a manuscript to the journal Buildings. It is up to the reviewer and the Journal Editor to judge whether or not the manuscript, thematically in its present form, falls within the scope of this journal.
Answer 2: the authors thank the reviewer for the explanation.
Reviewer's comment 3-Round 2: The authors use the terms 'geometry' and "topology." However, these terms refer to patterns or spatial elements - 3D, and since the pattern on the plate was printed and therefore also had a third dimension - depth, which I asked about and about which I found no mention in the text of the manuscript and the author's response to my comments. After all, the depth of the printed pattern also affects its surface area from a geometrical and topographical point of view and the water capacity of the printed pattern. Because it is a spatial pattern, the dept can also affect the amount of water held and the immobilization of algal strains in any orientation. In addition, there is no information as to the ratios of the area of the two patterns to the total area of the tiles, although the dimensions: of length, width, and thickness of the tiles are given.
Answer 3: Please read carefully the manuscript before claiming that there is missing information: the depth is expressed in Z offset that is stated clearly in the materials and methods section ¨3D Translation and 3D Printing of the Bioreceptive tiles: P1, and P2: Z-Offsetting, and Fractal Dimension: To translate the produced 2D reaction-diffusion patterns of the polar-periodic bioreceptive tile P1 and the strip-labyrinth bioreceptive tile P2, an XY-offsetting and Z- extrusion were applied to each of the two patterns, to provide spatiotemporal reaction-diffusion patterns translated into 3D topologies. The XY-offsetting was informed by the average scale lengths of the tested algal strains that are intended to be immobilized on these two bioreceptive tiles. This was determined by the identification of the cell size and morphology of each of the tested algal strains ranging from 25 µm to 1 cm. The offsets were converted to surfaces that were extruded in the Z direction with an equal numeric value to their width to create proportionate spatial topologies on each bioreceptive tile...¨ Furthermore, in the modified Figures 1,2,4,5 there are scale bars to show the ratio of the patterns to the overall dimension of the tile.
Reviewer's comment 4-Round 2: Then please write in the text of the manuscript that the study used "expressed tap water", but even this "expression tap water" had some mineral composition.
Answer 4. No, the authors meant by ¨expression of tap water¨ a usual expression and experimental procedure that is used widely in literature in the case of nonsynthetic tap water. the composition of tap water is by a negligible amount of minerals that don't affect the stock media constituents in algal culture media, thus, the authors do not agree with this comment as it is irrelevant or influential to the experimental methodology.
Reviewer's comment 5- Round 2: Interesting way to control the research, and where to find the results for comparison?
Answer 5: the authors are uncomfortable with the reviewer's comment that is clearly using irony. However, the answer to this comment is present in ¨Figure 3. which compares the growth rates and cell culture density compared to the initial culture density per strain. a) cell culture density of the activated cultures of the different algal strains after 4 weeks of cultivating each strain in its optimum growth media and conditions, compared to the starter culture density per strain. b) the estimated cell culture density per each immobilized algal strain on the bioreceptive tile P1, in comparison to the inoculum (activated) culture density per strain. and c) the estimated culture density per each immobilized algal strain on the bioreceptive tile P2, in comparison to the inoculum culture density per strain.¨ and its discussion in the results and discussion section. As it is clearly stated since the beginning compared to control groups were present at each step; by comparing first the initial separate cultures' growth to the growth of the activated culture in optimum media per each strain; then comparing each immobilized strain culture on the different bioreceptive tiles to these initial and activated cultures. The objective is to prove the efficiency of these specific two patterns in immobilizing the various algal strains by comparing them to the activated culture density which is the control group that shows the optimum growth conditions for each algal strain and hence the optimum growth ratio to compare to.
Reviewer's comment 6-Round 2: The required format applies to manuscripts at the level of submission and not acceptance. The reviewer assesses the content and form of the manuscript that will be published, not just the content, and the form will be reworked as the content is accepted. The fact that a journal editor decided to submit a manuscript for review in a form incompatible with the publisher's form can only be due to the editor's courtesy, but this does not mean that the reviewer's comments can be ignored.
Answer 6: This is an editorial aspect that it is not up to the authors to answer it. however, changing the format of the manuscript will definitely not change its content. And it can be easily done better after all amendments of addition or removal of sentences or paragraphs or other corrections are done to the manuscript.
Reviewer's comment 7- Round 2: According to the rules for responding to reviewer comments, authors should indicate the line numbers in the revised manuscript where they have made changes under the reviewer's comments.
Answer 7: Or use the Microsoft word editing and reviewing tracking system, which the authors have used and it exhibits clearly and more accurately any change that the authors have did to the manuscript following the reviewers´ comments. where lines´ numbers will not be accurate if there are newly added sentences or paragraphs or corrected figures as is the case in the current manuscript. in this case, line numbering will only be visually confusing, hard to find, and track changes done. please note that the editorial team themselves have instructed the authors to use the MS Word tracking system for tracking the modifications done to the manuscript. which implies that it is within the rules of responding to reviewer comments.
Additional comments: In addition, a misspelling of platelet sizes in the revised manuscript was noted (lines: 158 and 159), should be either: 15.0 cm x 15.0 cm x 0.5 cm or as the capacity resulting from the multiplication of the given values, i.e., 112.5 cm3. The notation "cm^3" is used as a command for calculations in an Excel spreadsheet and has nothing to do with the volumes and values presented in the manuscript's text.
Answer: Done, corrected.
Reviewer 2 Report
I think it can be accepted now.
Author Response
Thank you